# Socioeconomic context influences the heritability of child cortical structure
Linn B. Norbom [1] ✉, Espen M. Eilertsen [1], Andreas Dahl [2,3], Valerie Karl[1], Lars T. Westlye [2,3,4] &
Christian K. Tamnes [1,5]

Children differ in brain cortical morphometry and microstructure, which together form the structural foundation for cognition. Cortical structure is highly heritable, but whether heritability varies across socioeconomic status (SES) is unknown. In this preregistered study, we estimated single-nucleotide polymorphism (SNP)-based heritability of cortical thickness, surface area, sulcal depth, and grey-/white-matter contrast (GWC) among 9,080 US 10-year-olds. We then tested whether genetic and environmental contributions were moderated by parental SES, defined as a composite of income, education, and neighbourhood deprivation. Cortical thickness and surface area showed high heritability, while sulcal depth and GWC exhibited moderate heritability. However, among children from lower-SES backgrounds, cortical differences were less genetically related and more uniquely environmentally related, at times exceeding genetic contributions. These findings suggest that in contexts of socioeconomic disadvantage, children's brain structure reflect lived experience more strongly than previously recognized.

Nearly all human characteristics arise from a combination of genetic and environmental influences. Heritability quantifies the proportion of variation in a trait that is attributable to genetic differences, with estimates based on single nucleotide polymorphism (SNP) typically being lower than those derived from family studies[1]. The cerebral cortex, the brain's outer folded layer, shows individual differences in morphological features such as thickness, surface area, and folding patterns, as well as in microstructural properties reflected in magnetic resonance imaging (MRI)-based signal intensity. These structural variations are closely linked to developmental stage[2,3] and relate to cognitive functions[4–7] and mental health[8,9].

Neuroimaging studies document that brain structure is highly heritable, with family-based estimates in the range of 60–80%, with age-, metric-, and region-specific variation[10–12]. However, research has largely overlooked the notion that genetic influences may vary across socioeconomic contexts[13–16]. While socioeconomic status (SES) is related to cortical structure itself[17–20], it is unknown to what degree its genetic influences also depend on SES. Genetic moderation by SES has mainly been explored in cognitive research.

Twin studies estimate cognitive abilities to be 50–70% heritable, depending on age and the specific ability studied[21–24]. Several studies have found that the heritability of cognition is positively moderated by SES. This pattern, known as the "Scarr-Rowe hypothesis", was first reported in the US

in the 1970s[16] and later replicated in Sweden[25] and again in the US[15], though recent findings are mixed[26–28]. A commonly proposed mechanism for this relation is that children from low SES backgrounds have fewer opportunities to develop their predispositions, limiting the expression of their genetic potential. In contrast, genetic influences on cognitive traits may be max-imized in high SES-settings, where resources are more readily available[29,30]. Alternative theoretical frameworks include the "Pareto hypothesis" which posits that the genetic potential of cognitive abilities is most fully expressed in middle-class environments and restricted in both low and high SES groups due to limiting-, and buffering protective factors, respectively[31]. The "Saunders hypothesis" focuses solely on the protective role of high SES, suggesting that genetic potential is less fully expressed within this stratum[31].

Despite extensive research on how the heritability of cognitive abilities varies by socioeconomic standing, no study has examined whether SES similarly moderates the heritability of cortical structure. Addressing this gap could reveal whether children's brains are shaped differently by their inherited predispositions and lived experience, depending on socio-economic context.

In this preregistered study, we analysed cross-sectional data from over 9000 children and their parents, including 2544 children with at least one co-enrolled sibling, from the Adolescent Brain Cognitive Development (ABCD) Study. Genetic effects were assessed via SNP genotypes, child

[1]PROMENTA Research Center, Department of Psychology, University of Oslo, Oslo, Norway. [2]Department of Psychology, University of Oslo, Oslo, Norway. [3]Center for Precision Psychiatry, Division of Mental Health and Addiction, Oslo University Hospital, Oslo, Norway. [4]K.G Jebsen Center for Neurodevelopmental Disorders, University of Oslo, Oslo, Norway. [5]Division of Mental Health and Substance Abuse, Diakonhjemmet Hospital, Oslo, Norway.
✉e-mail: l.c.b.norbom@psykologi.uio.no

**Fig. 1 | Lobar heritability estimates for cortical thickness, surface area, sulcal depth, and grey/white-matter contrast.** This figure presents lobar heritability estimates for cortical thickness, surface area, and grey/white-matter contrast, mapped onto the Desikan–Killiany atlas. The left and right hemispheres are identical within each metric due to averaging across hemispheres.

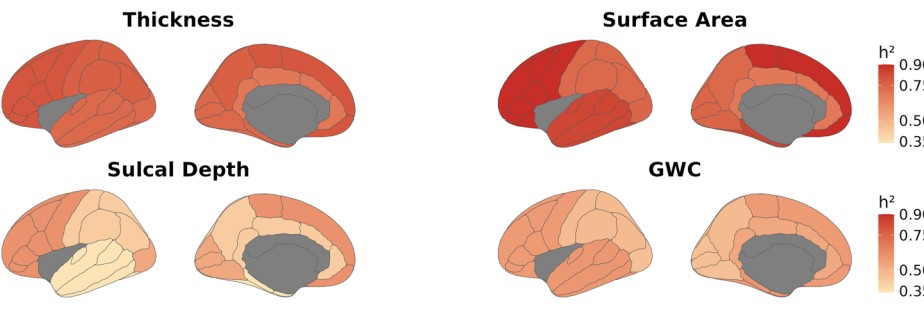

cortical structure via MRI based global and lobar cortical thickness, surface area, sulcal depth, and grey/white-matter contrast (GWC), and SES via a composite of parental income, education, and neighbourhood deprivation. We first estimated the overall heritability ($h^2 = A/(A + C + E)$) of cortical structure, where A is the additive genetic variance, C is the environmental variance shared by siblings, and E is the environmental variance unique to the child. Because $h^2$ is a *proportion of total variance* attributable to genetics, it depends on the denominator; for example, increasing environmental variance (C or E) lowers $h^2$ even if A is unchanged. Next, we tested whether the absolute variances A, C, and E differed across socioeconomic contexts. Additionally, we examined the genetic and environmental effects on cognitive abilities within the same sample.

## Results
### Genetic contribution to cortical structure and moderation by parental SES

**Cortical thickness.** Baseline linear mixed effects modelling without genetic or environmental interaction terms revealed a SNP-based heritability for mean cortical thickness of $h^2 = 0.82$ after controlling for age, sex, and genetic principal components (PC) related to population stratification.

Likelihood ratio tests comparing the baseline main effects-model and the interaction model between the SES index and additive genetic, shared environment, and unique environmental factors yielded a non-significant interaction after multiple comparisons correction ($x^2 = 8.27$, Df = 3, $p = 0.04$, corrected $p = 0.051$), thus providing no indication that the heritability of global cortical thickness differs across SES levels. Baseline lobar analyses revealed that the heritability of cortical thickness varied between 0.69 and 0.80, depending on region (Fig. 1 and Supplementary Information (SI) Table 1). Further, likelihood ratio tests indicated that the genetic and environmental factors contributing to the heritability of parietal ($x^2 = 12.56$, Df = 3, $p = 0.006$, corrected $p = 0.015$) and occipital thickness ($x^2 = 9.47$, Df = 3, $p = 0.024$, corrected $p = 0.04$) varied across SES levels (Table 1). Specifically, the heritability of parietal and occipital thickness was higher for children in high SES settings, due to increased additive genetic effects and reduced unique environmental influences, while shared environmental effects remain consistently negligible across SES levels (Fig. 2).

Heritability estimates for all imaging metrics, whether derived from a more genetically homogeneous subsample (SI Figs. 5–6, SI Table 5) or obtained using a twin-based method, yielded similar results, are presented in the SI Results (SI Tables 2, 3).

**Surface area.** SNP-based heritability for total surface area was $h^2 = 0.82$ after controlling for age, sex, and population stratification. Likelihood ratio tests yielded a non-significant interaction between the SES index and genetic and environmental factors after correction ($x^2 = 8.34$, Df = 3, $p = 0.04$, corrected $p = 0.051$), thus providing no indication that the heritability of global surface area differs across SES levels. Lobar analyses revealed that heritability varied between 0.69 and 0.90 (Fig. 1 and SI Table 1). Moreover, genetic and environmental contributions to 4 out of 5 lobes, namely frontal, parietal, temporal, and occipital surface area,

varied across SES (Table 1). Specifically, the heritability of surface area was higher for children from high-SES settings due to increased additive genetic effects, while environmental influences generally remained negligible-, or were lower in high SES settings. This was except unique environmental influences, which were slightly higher in high-SES settings for occipital surface area (Fig. 3).

**Sulcal depth.** Mean sulcal depth showed a heritability of $h^2 = 0.57$ after controlling for relevant covariates. There was no significant interaction between the SES-index and the genetic and environmental factors ($x^2 = 1.08$, Df = 3, $p = 0.781$, corrected $p = 0.781$), thus providing no indication that the heritability of global sulcal depth differs across SES levels. Lobar heritability varied between 0.34 and 0.61 (Fig. 1 and SI Table 1). Due to the low chi-square value and non-significance before correction, we did not proceed with lobar assessments of sulcal depth.

**GWC.** SNP-based heritability for GWC was $h^2 = 0.57$ after controlling for age, sex, and population stratification. Likelihood ratio tests revealed an interaction between the SES index and genetic and environmental factors ($x^2 = 13.59$, Df = 3, $p = 0.004$, corrected $p = 0.010$). The results indicate that the heritability of global GWC was higher in children from high-SES backgrounds due to increased additive genetic effects and a marked reduction in unique environmental influences, along with a slight decrease in shared environmental effects. In contrast, for children from low-SES backgrounds, unique environmental contributions exceed genetic contributions, accounting for a greater proportion of the variance in GWC under these circumstances (Fig. 4). Lobar analyses showed that the heritability of GWC varied between 0.48 and 0.59 (Fig. 1 and SI Table 1). Genetic and environmental factors varied across SES levels for four out of five lobes, namely the cingulate, frontal, parietal, and occipital lobes (Table 1), with patterns generally resembling the global pattern (Fig. 5), except for a somewhat varying contribution from shared environmental influences.

### Genetic and environmental influence on general cognitive ability and moderation by parental SES

Baseline linear mixed effects modelling without genetic or environmental interaction terms revealed a SNP-based heritability for general cognitive ability of $h^2 = 0.57$ after controlling for age, sex, and population stratification. Likelihood ratio tests comparing the baseline main effects-model and the interaction model between the SES index and additive genetic, shared environment, and unique environmental factors yielded a significant interaction ($x^2 = 29.60$, Df = 3, $p$ and corrected $p \leq 0.001$), indicating that these factors, which constitute the heritability of general cognition, vary across SES levels. More specifically, the heritability of general cognition was higher for children from high-SES settings, due to increased additive genetic effects and a reduction in unique and, to a lesser extent, shared environmental influences. In contrast, for children from lower SES, genetic and environmental impact appeared similar, with unique environmental influences exceeding genetic contributions (Fig. 6).

## Discussion

In a sample of over 9000 10-year-old children from the US, we found that differences in cortical thickness and surface area were primarily accounted for by genetic variation, while cortical folding and intensity showed moderate heritability. Across cortical metrics, the frontal lobe was the most heritable region. These estimates align with previous research[32–35], yet our study shows that they do not fully generalize to children from lower socioeconomic backgrounds, where regional cortical structure appears to be more related to unique environmental factors, at times exceeding genetic contributions. This parallels findings for cognitive abilities, including our own, and suggests that the balance between genetic and environmental contributions to child brain structure also shifts with socioeconomic context.

We find that cortical thickness and surface area are about 80% heritable in childhood, in line with general heritability estimates for T1-weighted cortical structure across the lifespan[33]. Our estimates show some discrepancies with those reported in a previous study using twin-based methods on the same sample ($n = 1544$)[36]. The study reported global thickness and surface area estimates of 66% and 95%, respectively, but with lower and more varied ROI based estimates. Discrepancies between the studies include their adjustment for self-reported ethnicity and three twin sites, whereas we included SES and genetic PCs in our models and harmonized cortical data across scanners. Leveraging a Canadian dataset of 57 monozygotic and 35 dizygotic 8-year-old twins, Yoon et al.[37] reported vertex-wise heritability estimates of cortical thickness ranging from 0 to a maximum of 78% (left) and 67% (right)[37]. Assessing the twins' surface area in native imaging space resulted in interclass correlations that indicated left-to-right heritability asymmetries of approximately 62% and 46% respectively (Yoon et al.[37]). However, these studies had a more limited sample size.

We found that sulcal depth and GWC were both moderately heritable, with estimates approaching 60%. As a pure intensity metric, GWC is more susceptible to noise than geometry-based morphometry, potentially lowering the heritability estimate. Still, while no prior study has specifically assessed the heritability of either metric in childhood, our results are generally in line with previous studies of related metrics or in young adulthood. Maes and colleagues[36] reported average convexity heritability below 40% across most ROIs. In young adulthood, Pizzagalli et al.[11] estimated heritability to be 40% for mean sulcal depth in two separate Australian and American samples[11], and Schmitt et al.[38] reported heritability estimates of 49% for average convexity and 61% for mean curvature in the same American sample. A study of Vietnamese twins in midlife reported ROI based heritability for GWC ranging from 0 to 66%[34], and a US-based twin study found approximately 50% heritability for a related intensity metric, the T1-weighted/T2-weighted (T1w/T2w)-ratio[39].

### Table 1 | Interaction tests for relevant imaging metrics

| Imaging metric and lobe | ChiSq | Df | P | P corrected |
|---|---|---|---|---|
| Thickness cingulate | 0.11 | 3 | 0.991 | 0.991 |
| Thickness frontal | 4.05 | 3 | 0.256 | 0.295 |
| Thickness occipital | 9.47 | 3 | **0.024** | **0.040** |
| Thickness parietal | 12.56 | 3 | **0.006** | **0.015** |
| Thickness temporal | 6.61 | 3 | 0.086 | 0.114 |
| Area cingulate | 6.48 | 3 | 0.091 | 0.114 |
| Area frontal | 9.71 | 3 | **0.021** | **0.039** |
| Area occipital | 8.73 | 3 | **0.033** | **0.050** |
| Area parietal | 11.54 | 3 | **0.009** | **0.019** |
| Area temporal | 12.91 | 3 | **0.005** | **0.015** |
| GWC cingulate | 16.22 | 3 | **0.001** | **0.007** |
| GWC frontal | 13.00 | 3 | **0.005** | **0.015** |
| GWC occipital | 3.55 | 3 | 0.315 | 0.337 |
| GWC parietal | 12.30 | 3 | **0.006** | **0.015** |
| GWC temporal | 16.95 | 3 | **0.001** | **0.007** |

The table presents interaction test results for various imaging metrics across brain lobes. It includes Chi-square (ChiSq) values, degrees of freedom (Df), uncorrected (*P*), and corrected *p*-values (*P* corrected). Significant interactions (*p* ≤ 0.05) are marked in bold. Based on global findings, lobar sulcal depth was not tested.

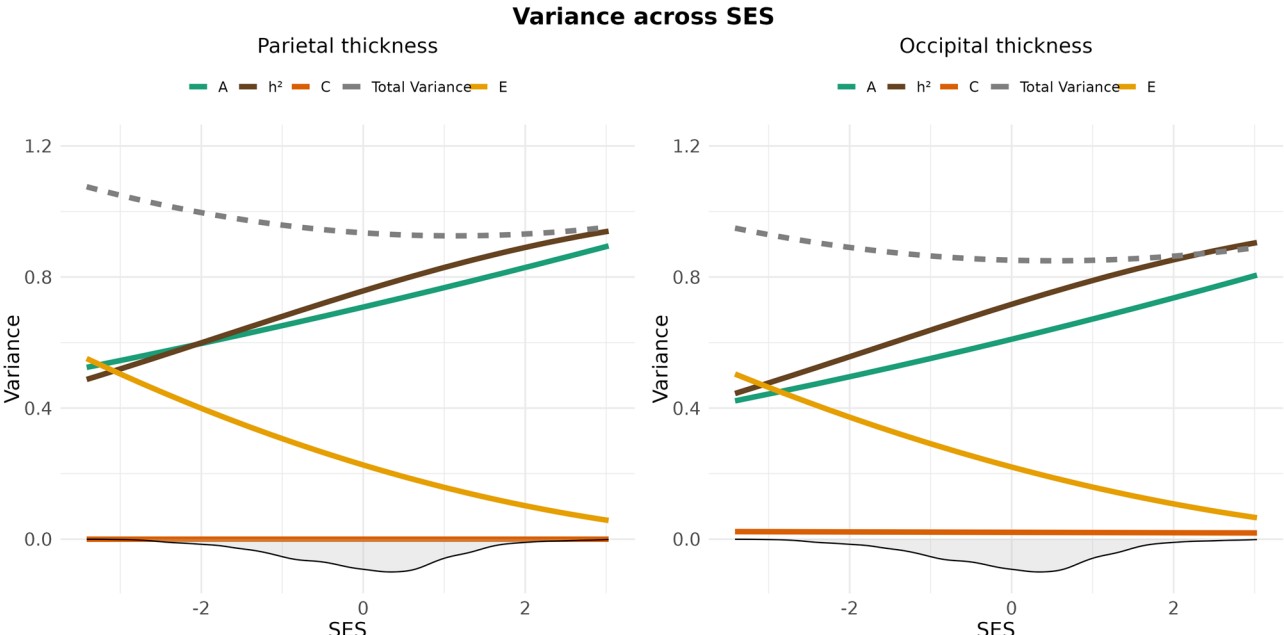

**Variance across SES**

**Fig. 2 | Genetic and environmental variance of parietal (left) and occipital (right) thickness across SES.** The image displays splines representing heritability ($h^2$ = A/ (A + C + E); brown), along with its components: additive genetics (A; green), shared environment (C; dark orange), and unique environment (E; yellow) for cortical thickness across the standardized SES construct. The density distribution of SES is shown as a shaded area below the *x*-axis. The dotted grey line shows the total phenotypic variance (A + C + E).

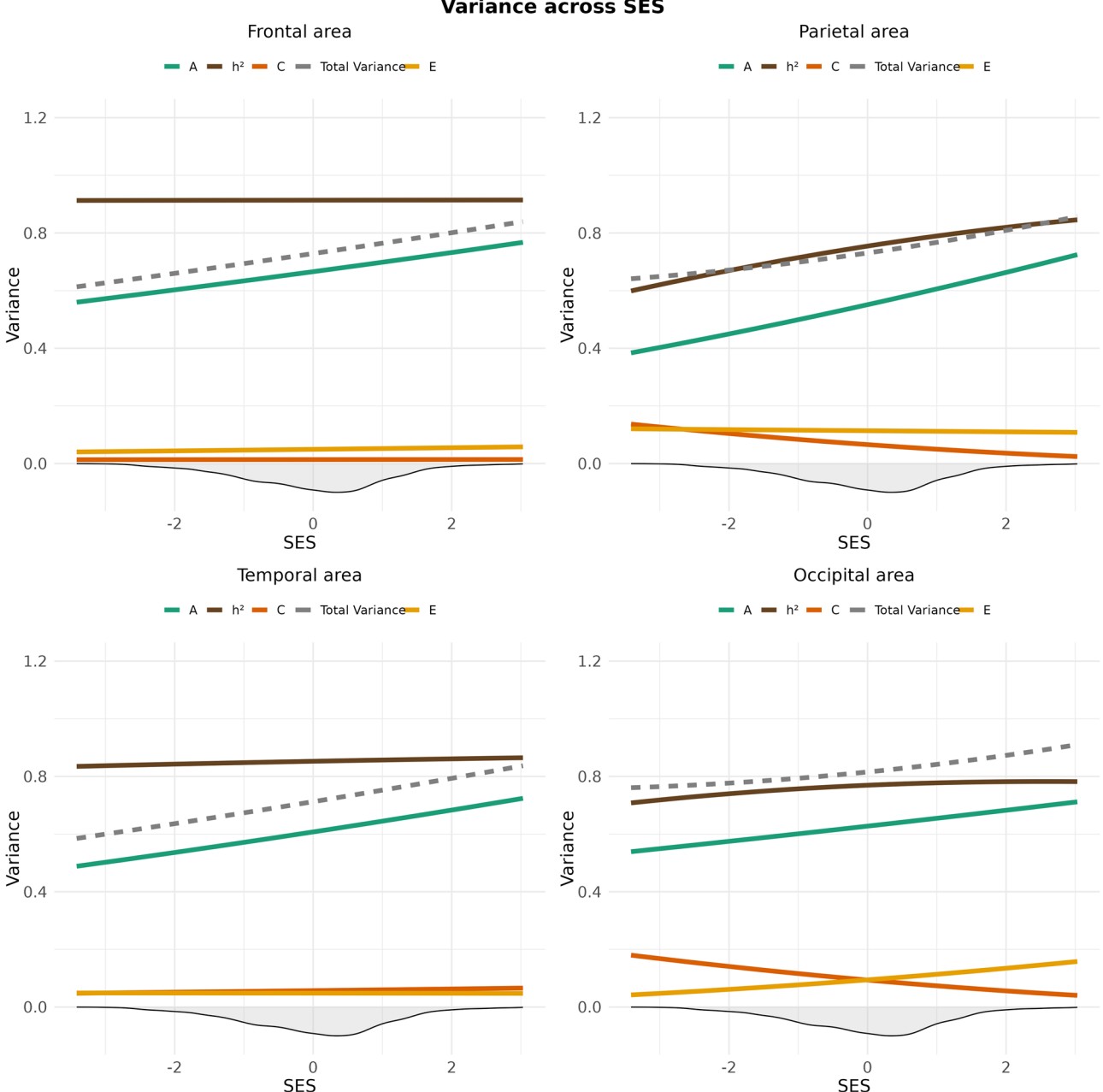

**Fig. 3 | Genetic and environmental variance of frontal (upper left), parietal (upper right), temporal (lower left) and occipital (lower right) surface area across SES.** The image displays splines representing heritability ($h^2 = A/(A + C + E)$; brown), along with its components: additive genetics (A; green), shared environment (C; dark orange) and unique environment (E; yellow) for cortical surface area across the standardized SES construct. The density distribution of SES is shown as a shaded area below the x-axis. The dotted grey line shows the total phenotypic variance (A + C + E).

The heritability of cortical morphometry and microstructure is lower in children from lower- compared to higher SES families. This pattern was evident for occipital and parietal thickness, as well as for most of lobar surface area and GWC, but not for cortical folding. While genetic differences accounted for most of cortical variation among children from high-SES backgrounds, in low-SES settings, unique environmental factors appeared to play a larger role, corresponding to-, or even exceeding the contribution of genetics. Shared environment, or specifically, being part of the same family, was minimally associated with cortical structure in our study. Both the inclusion of families with more than two participating siblings and the use of molecular genetic data help to better separate shared genetic from shared environmental effects[40]. The small shared environment contribution could still reflect limited statistical power, as only 28% of participants had enrolled siblings in the dataset, most of whom were from two-sibling families. The

moderating effects of SES on the heritability, or on the genetic and environmental components, of cortical structure have not been previously tested. However, Chiang et al.[41] used 705 Australian twins and their siblings to assess whether the heritability and additive genetic contributions on white matter fractional anisotropy were moderated by occupational status. While they found no moderating effect on the heritability estimate, occupational status interacted with the additive genetic component in isolated white matter voxels. Our results suggest that, much like previous (albeit inconsistent) findings for cognitive abilities[15,27,28,30], and in line with our own cognitive findings, the genetic and environmental contributions to regional cortical morphometry and microstructure, key structural correlates of cognition, also vary with SES.

Our study cannot determine why unique environmental conditions associated with socioeconomic disadvantage play a greater role in explaining

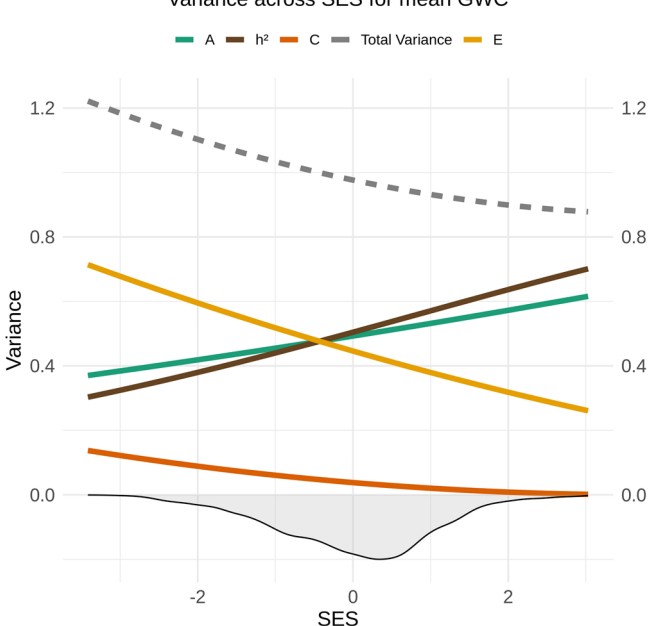

**Fig. 4 | Genetic and environmental variance of mean grey-white matter contrast (GWC) across SES.** The image displays splines representing heritability ($h^2$ = A/(A + C + E); brown), along with its components: additive genetics (A; green), shared environment (C; dark orange), and unique environment (E; yellow) for mean GWC across the standardized SES construct. The density distribution of SES is shown as a shaded area below the $x$-axis. The dotted grey line shows the total phenotypic variance (A + C + E).

additive genetic effects such as dominance or epistasis. Finally, unique environmental influences are not specifically modelled, and as such, we cannot probe which experiences or exposures underlie the non-shared environmental variance.

Children differ in their cortical brain architecture. Our findings show that genetic and environmental contributions to regional cortical thickness, most of surface area, and both global and regional microstructure vary between socioeconomically advantaged and disadvantaged children. Among socioeconomically advantaged children, inherited predispositions appear to play a larger role in brain structural differences, whereas in contexts of socioeconomic disadvantage, structural brain variation may more strongly reflect a child's lived experience. Our study underscores the importance of considering context when interpreting brain heritability estimates.

## Methods and materials
### Participants
Data was acquired from the ABCD Study using the annual release 5.1 (https://data-dict.abcdstudy.org/?) and 3.0 for SNP-based information. The ABCD study consists of nearly 12,000 US children, approximately 9–10 years old at the time of study inclusion, as well as their parents, with planned follow-ups for 10 years. The protocol contains comprehensive assessments of demographic information, genetics, behavior, and brain structure and function[49]. Recruitment was generally conducted through the school system and was informed by sex, ethnicity, SES, and urbanicity. Parental informed consent as well as child assent were obtained for all participants. The Institutional Review Board at the University of California, San Diego approved all aspects of the ABCD Study[50], and the current study was conducted in line with the Helsinki Declaration and approved by the Regional Committee for Medical and Health Research Ethics (REK 2019/943). All ethical regulations relevant to human research participants were followed. Exclusion criteria and further details are presented in SI Methods.

The current study focused on baseline assessments from 11,868 participants. 1934 individuals were excluded due to missing data on either initial demographic, socioeconomic, neighborhood, cognitive, genetic ethnicity, family ID, scanner, imaging quality, or cortical macro- or microstructure-based variables of interest (SI Fig. 1). 213 individuals were excluded due to failing imaging-based quality control (QC) (described below), and 641 children were excluded due to missing or poor-quality SNP-based relatedness data. SI Fig. 2 shows the spread of SES and cognitive abilities for excluded individuals. The final sample thus consisted of 9080 children (4266 = females) aged 8.9–11.1 years (mean = 9.9 SD = 0.63). Two individuals who were assigned intersex-male at birth were recoded as male to enhance statistical robustness while ensuring the inclusion of all relevant individuals to our study. The sample comprised 6536 children from singleton families and 2544 children from sibship families, with counts reflecting enrolled siblings rather than the actual household size. Of these, 2428 children came from two-sibling families, 111 from three-sibling families, and five from a five-sibling family. The dataset included 670 twin pairs—261 monozygotic and 409 dizygotic—plus 7 complete triplet sets and 1 incomplete set in which only two of the three siblings met inclusion criteria.

### Measurements of socioeconomic status and neighbourhood deprivation
Socioeconomic information was reported by a parent or guardian on behalf of themselves and a partner if relevant. Parental and partner education was assessed by indicating the highest grade level or degree received, on a 22-point scale. We recoded this scale to assimilate years of education (SI Table 4), and parental education was then defined as the highest educational score of either the reporting parent or their partner if applicable.

Parental and partner income was assessed by indicating their earnings before taxes and other deductions over the past 12 months on a 10-point scale. Total combined income was measured using the same scale, reflecting all sources of family income over the past 12 months. We recoded these

differences in child cortical structure. While our finding of positive SES moderation aligns with the Scarr-Rowe hypothesis, the underlying mechanism proposed by Scarr-Rowe may not. In our data, overall phenotypic variance in most cortical and general cognitive metrics was larger in lower-SES strata, suggesting that children from high-SES backgrounds are more similar to one another. Surface area, however, showed the opposite pattern. Still, if this were to generally reflect greater environmental homogeneity, it might align more closely with the "Saunders hypothesis"[31] than with the individualized opportunities posited for high-SES contexts in the Scarr-Rowe framework[29,30]. Based on the overall pattern observed in our data, although speculative, it is also conceivable that in high SES environments, neurobiological processes, which are under strong genetic control[42,43] and assumed to influence MRI based cortical structure, such as dendritic arborization, axon caliber, and cortical myelin[3], cause differences in cortical structure. In contrast, in children from low SES families, certain unique environmental events or exposures (adverse or other) are more frequent and have broad non-specific effects on the same neurobiological processes, thus becoming the primary driver of cortical differences in these children. This is somewhat similar to the "social push" hypothesis, which posits that environmental as opposed to genetic factors become more central in explaining outcomes when conditions are extreme or adverse[44].

There are several limitations to our study. First, although we aimed to create a broad SES construct, we were unable to reliably model the key subfactor, parental occupation. Occupation shows moderate correlations with income and education[17,18], and captures *symbolic capital* or prestige[45] and work-environmental conditions linked to physical and mental health[46].

Second, the index is not sensitive to family size ("income-to-needs") or geographic variation in cost of living across the US, and does not capture subjective, relative, or cultural aspects of SES[47,48]. Third, our study can only capture associations and cannot infer causation. Third, our analyses are based on SNP data within families and therefore do not account for rare genetic variants, which, in some cases, have profound impact on both brain structure and function. Fourth, our ACE model cannot capture non-

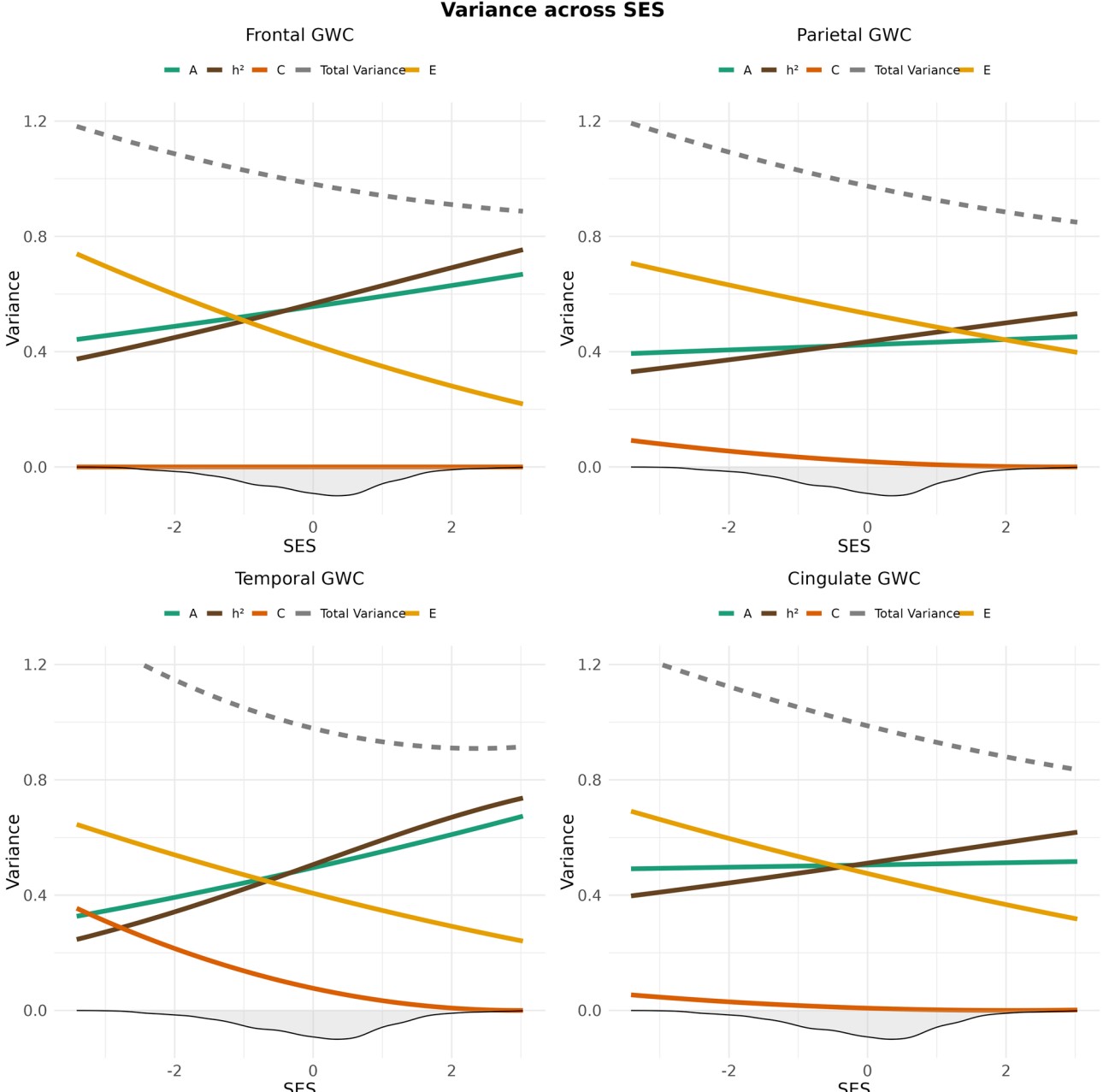

**Fig. 5 | Genetic and environmental variance of frontal (upper left), parietal (upper right), temporal (lower left) and cingulate (lower right) GWC across SES.** The image displays splines representing heritability ($h^2 = A/(A + C + E)$; brown), along with its components: additive genetics (A; green), shared environment (C; dark orange), and unique environment (E; yellow) for GWC across the standardized SES construct. The density distribution of SES is shown as a shaded area below the *x*-axis. The dotted grey line shows the total phenotypic variance (A + C + E).

income brackets to the median point, pragmatically setting the maximum of the top bracket to $350,000 (SI Table 4) based on online reports indicating that the threshold for the top 1% of U.S households in 2018 was $434,454.80 (https://dqydj.com/2018-average-median-top-household-income-percentiles/). Parental income was defined as the highest value reported from parent, partner, or combined income. To account for diminishing marginal utility, we applied a natural logarithm transformation to the income data.

Neighborhood deprivation was assessed using a composite score from the Area Deprivation Index (ADI) in the "Residential History Derived Scores" data file. The ADI is the scaled and weighted sum of 17 US census variables, including measures of poverty, education, housing, and employment, with a higher score indicating greater deprivation[51].

In line with our pre-registration, we did not include assessments of occupation in this study. The rationale, detailed in the SI Information, is that the ABCD Occupation Survey does not rank occupations by status but instead provides status-neutral categories.

To create a single overarching SES index, we reduced z-scored parental education, parental income, and neighborhood deprivation by principal component analysis (PCA) in R using the "prcomp" package. PCA was chosen as it is a conventional data driven reduction that avoids a priori assumptions that SES-subfactors contribute equally or with preset weights and maximizes shared variance while remaining sensitive to noise differences. The first component, which explained 65% of the total variance, was extracted as the overarching "SES-index". It correlated strongly with parental education ($r = 0.84$), parental income ($r = 0.87$), and neighborhood

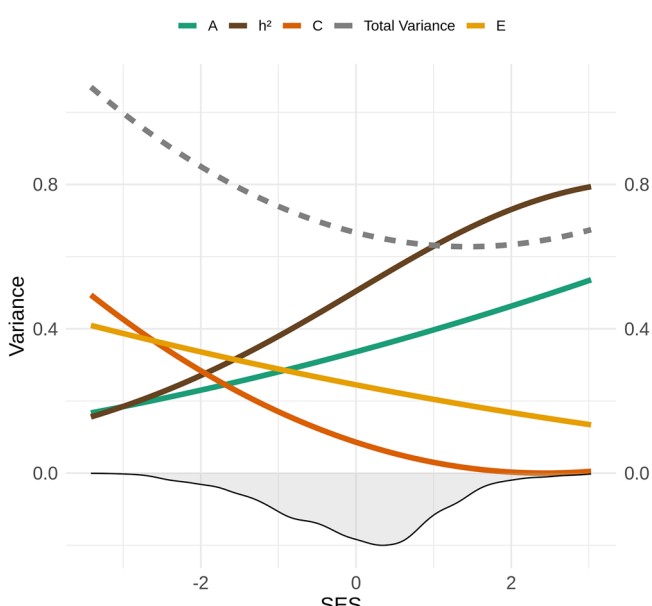

**Fig. 6 | Genetic and environmental variance of general cognitive ability across SES.** The image displays splines representing heritability ($h^2 = A/ (A + C + E)$; brown), along with its components: additive genetics (A; green), shared environment (C; dark orange), and unique environment (E; yellow) for general cognitive ability across the standardized SES construct. The density distribution of SES is shown as a shaded area below the *x*-axis. The dotted grey line shows the total phenotypic variance (A + C + E).

deprivation (r = –0.71). Higher values reflect higher income and education, and lower neighborhood deprivation (SI Fig. 3). See SI Fig. 4 for the distribution of SES and neighborhood-related variables in the final sample.

*Genotype Data Processing and Relatedness Estimation.* Genotype data were based on Data Release 3.0 and derived from saliva and blood samples[52]. Genotyping was conducted using the Affymetrix NIDA SmokeScreen Array[53], containing 733,293 SNPs, with samples processed at Rutgers University Cell and DNA Repository (RUCDR). Initial DNA-based QC was performed by RUCDR, followed by further QC by the ABCD Data Analysis and Informatics Resource Center (DAIRC) using the Ricopili pipeline[54]. Pre-imputation steps followed TOPMed documentation and included identifying rare and common variants with PLINK v1.9, ensuring format consistency, converting to VCF format, and uploading to the TOPMed Imputation Server[55]. Data was then imputed, with mixed ancestry references to improve accuracy for diverse populations, and phased with Eagle v2.4 to determine allele arrangement on each chromosome.

Genetic relatedness among participants was estimated from the imputed genotype data[55]. The matrix of all pairwise genetic relatedness coefficients is referred to as the GRM, with each cell representing genetic similarity between pairs and self-relatedness along the diagonal. In samples of unrelated individuals, this matrix can be used to estimate the additive genetic contribution of all tagged and imputed SNPs[1]. When applied to samples of related individuals, Smith and colleagues[24] suggested that all relatedness coefficients in the GRM could be set to zero between non-family members, such that the additive genetic effect is only correlated among members of the same family. The method was suggested to perform well when applied to cognitive phenotypes in the ABCD study, and was implemented in the current application.

For validation, we additionally calculated heritability using Falconer's formula (see SI Methods). This traditional twin-based analysis was not pre-registered and used only as a supplementary check.

## MRI acquisition, quality control, preprocessing, and scanner harmonization

Imaging data was acquired on 29 separate 3T scanners from Siemens Prisma, General Electric (GE) 750, and Philips. The T1w image was an inversion-prepared RF-spoiled gradient echo scan, with prospective motion correction when available, and a 1 mm isotropic voxel resolution. Detailed descriptions of acquisition parameters and child-friendly MRI practices are presented elsewhere[56].

Raw QC of the T1w sequences was performed by the ABCD Data Analysis and Informatics Core. They used a standardized pipeline of automated and manual procedures that yielded a binary code for images recommended for inclusion[57].

We relied on FreeSurfer version 7.1.1 runs provided by the ABCD consortium to perform volumetric segmentations and cortical surface reconstructions. This process includes defining the "white" and "pial" surface, which is the grey/white matter boundary and the grey/cerebrospinal fluid (CSF) boundary, respectively[58,59]. Cortical thickness is then computed as the shortest vertex-wise distance between the white and pial surface, while vertex-wise surface area is calculated by summing the triangle areas converging at each particular vertex on the white surface. Vertex-wise-sulcal depth is calculated as the mm signed distance a vertex moves during an inflation operation from the white- to the inflated surface. Values are median-standardized and scaled based on variance, so that a positive value indicates outward motion more than the median of all vertices, indicating a deeper sulcus, whereas a negative value indicates inward motion and shallower sulci. GWC is calculated using intensity sampling from "rawavg.mgz", where white matter is sampled 1 mm below-, and grey matter is sampled 30% above the white surface. The vertex-wise percentage is then computed as:

$$100 \times \frac{white - grey}{\frac{white+grey}{2}}$$

In this way, lower GWC reflects more similar grey and white matter, a blurring that is documented across youth development[9]. We used the global surface metrics from the tabulated files provided by the ABCD consortium, which are derived by averaging the values across all vertices, except for surface area, where values are summed into a total.

To examine the spatial variability of interactions between the SES index and $h^2$, we also performed analyses at lobal granularity (frontal, parietal, temporal, occipital, and cingulate), reducing the Desikan atlas across hemispheres following FreeSurfer recommendations[60]. The fusiform gyrus, which spans both the occipital and temporal lobes, was assigned to the temporal lobe due to its stronger association with higher-order visual functions. The insula, which does not map cleanly to any conventional lobe and was represented by single values per hemisphere, was not included in lobar analyses. Surface area was summed, while the remaining metrics were averaged using a weighted-mean approach based on surface area.

In addition to raw image QC, we excluded individuals whose original surface showed extreme topological defects (i.e., holes, bumps, or handles), defined as exceeding 4 standard deviations from the mean on the "defect topology" variable for the left and right hemispheres.

To adjust for systematic and unwanted scanner-related variance, imaging metrics were subsequently imported to R, and the package neuroCombat[61] was employed to harmonize data across each of the 29 scanners. We included relevant covariates to our model, namely age, sex, parental education, parental income, neighborhood deprivation, general cognitive ability, family ID, and 32 genetic PCs, to preserve such variance during the harmonization procedure. Box plots of global MRI measures pre- and post-neuroCombat adjustments are presented in SI Fig. 7–10.

## Measurement of general cognitive ability
Cognitive abilities were assessed using the neurocognitive battery "the NIH toolbox"[62–64], administered on an iPad while monitored by a research assistant. The battery comprises seven tasks, covering episodic memory,

**Fig. 7 | Pearson's correlations between relevant variables.** The figure shows a correlation matrix between age, the SES-index, general cognitive abilities, and global metrics after scanner harmonization for the final sample.

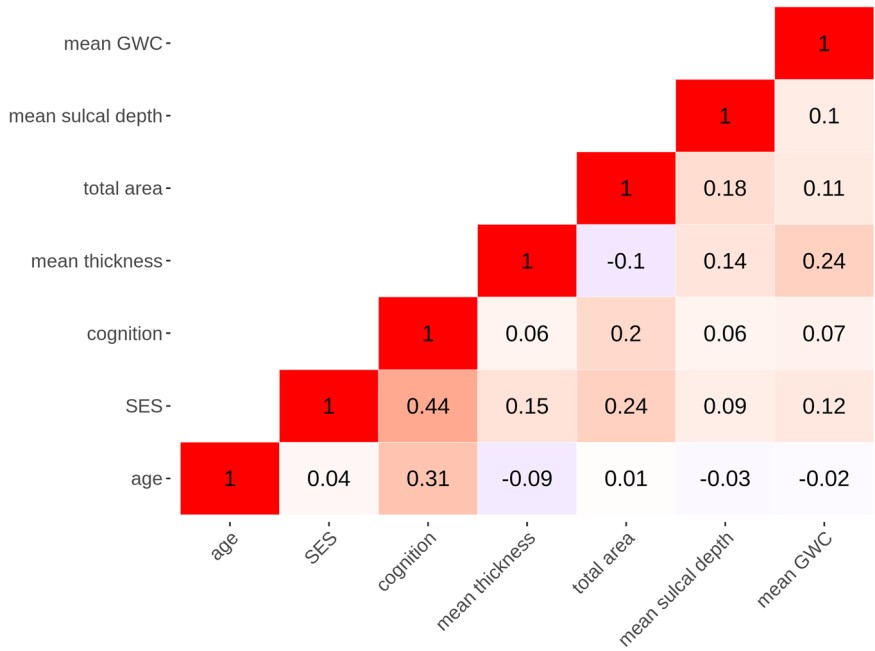

executive function, attention, working memory, processing speed, and language[65,66]. As a measure of general cognitive ability, we used the uncorrected total composite score of cognition. Figure 7 shows correlations between age, the SES-index, general cognitive ability, and global imaging metrics.

## Statistics and reproducibility

This study was pre-registered, where full statistical code is also available[67]. For the final sample ($n = 9080$), we fit a mixed effects "baseline model" for each imaging metric (cortical thickness, surface area, sulcal depth, and grey/white-matter contrast) and for general cognitive ability, using the svcm package in R[68]. These models, for individual $i$ in family $j$, include all main effects but no interaction terms and were specified as:

$$y_{ij} = w'_{ij}\gamma + \beta s_j + a_{ij} + c_j + e_{ij}.$$

In this linear model, ($y$) represented the cortical or cognitive metric, $\gamma$ was the fixed effects of the covariates (e.g., age, se,x and 32 genetic PCs derived by Fan et al.[55], related to minimizing population stratification), and $\beta$ was the fixed effect of the SES index ($s$). The term $a$ was a person-varying random additive genetic effect, distributed across individuals according to the GRM with variance $\sigma_a^2$. For example, for siblings $k$ and $l$, the GRM at position kl has a value of $\approx\frac{1}{2}$, reflecting their estimated sharing of genetic material. If the siblings are monozygotic twins, the value is $\approx 1$. These values represent the expected correlation in phenotypes that arise because of additive genetic effects. Of note, other sources of variation may lead to similar patterns, for example, if monozygotic twins are treated more similar by their parents. The term $c$ was a family-varying random effect, independent across families with variance $\sigma_c^2$. However, other sources that increase resemblance among relatives may lead to similar patterns, for example, non-additive genetic effects or assortative mating. The term $e$ was a person-varying random environmental/residual effect independent across individuals with variance $\sigma_e^2$. This term has the same structure as the residual in other linear models and captures other sources of variability that are not correlated across individuals, such as classical measurement error. Under this model, the genetic and environmental components of variance do not depend on the SES index. Of note, the preregistration detailed a genetic (a) + non-genetic (e) model, but with extensive family membership data available, we chose to further parse the non-genetic signal by also modelling the

variance shared between family members (c). Heritability ($h^2$) was calculated as the proportion of total variance, conditional on the fixed effects, explained by genetics:

$$h^2 \frac{\sigma_a^2}{\sigma_a^2 + \sigma_c^2 + \sigma_e^2}.$$

In a secondary set of "interaction models", we expanded the baseline model to include interaction terms between the SES index and random effects:

$$y_{ij} = w'_{ij}\gamma + \beta s_j + a_{1ij} + c_{1j} + e_{1ij} + s_j a_{2ij} + s_j c_{2j} + s_j e_{2ij}.$$

The three new terms $s_j a_{2ij} + s_j c_{2j} + s_j e_{2ij}$ were random slopes of $s_j$ regarding the genetic and environmental background factors, which had the consequence of inducing heteroscedastic variance. The random slope terms had similar distributions across individuals as in the baseline model, but with variances $\sigma_{a2}^2$, $\sigma_{c2}^2$, and $\sigma_{e2}^2$. Importantly, the genetic and environmental components of variance are now quadratic functions of SES. For the additive genetic variance, the function was:

$$\sigma_a^2(s) = \sigma_{a1}^2 + 2\sigma_{a1}\sigma_{a2}s + \sigma_{a2}^2 s^2.$$

The shared and unique environmental variance had similar functions with respect to the SES-index. This specification of the models is similar to what Purcell[69] referred to as "quantitative interactions". Consequently, the heritability cannot be evaluated without reference to the SES-index:

$$h^2(s) = \frac{\sigma_a^2(s)}{\sigma_a^2(s) + \sigma_c^2(s) + \sigma_e^2(s)}.$$

The baseline models were nested in the interaction models, and we obtained joint tests for interaction terms with likelihood ratio tests. All hypothesis tests were two-sided. Adjustments for multiple comparisons were performed using the Benjamini-Hochberg procedure for false discovery rate (FDR), separately for global and lobar analyses. Statistical significance was defined as FDR-adjusted $p \leq 0.05$. Of note, we left factors associated with SES and brain structure (e.g., pubertal timing, early-life trauma, and pre-/perinatal risk) unadjusted by design, as they relate to the

SES context of interest. Future mechanism-focused studies should model them explicitly.

Declaration of generative AI and AI-assisted technologies in the writing process: The manuscript was written by the corresponding author in collaboration with the co-authors. ChatGPT 4.0 was used to refine the English language and improve readability. After using this tool, the corresponding author reviewed and edited the content as needed takes full responsibility for all content within the manuscript.

## Reporting summary

Further information on research design is available in the Nature Portfolio Reporting Summary linked to this article.

## Data availability

Data were acquired from the ABCD Study via the NIMH Data Archive (NDA) using annual release 5.1 (https://data-dict.abcdstudy.org/) and release 3.0 for SNP-based information, under controlled access (Data Use Certification). Individual-level data are not publicly available; eligible researchers can apply for access through the NDA.

## Code availability

This study was pre-registered at https://osf.io/azucp, where the full statistical code is also available.

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

## Acknowledgements

Data was obtained from the Adolescent Brain Cognitive Development (ABCD) Study (https://abcdstudy.org), which is supported by the National Institutes of Health and additional federal partners under award numbers: U01DA041048,U01DA050989, U01DA051016,U01DA041022,U01DA051018,U01DA051037, U01DA050987,U01DA041174,U01DA041106,U01DA041117, U01DA041028,U01DA041134,U01DA050988,U01DA051039,U01DA041156,U01DA041025,U01DA041120,U01DA051038, U01DA041148,U01DA041093,U01DA041089,U24DA041123, U24DA041147. A full list of supporters is available at https:// abcdstudy.org/federal-partners.html. A listing of participating sites and study investigators can be found at https://abcdstudy.org/ consortium_members/. ABCD consortium investigators designed and implemented the study and/or provided data but did not necessarily participate in the analysis or writing of this report. This manuscript reflects the views of the authors and may not reflect the opinions or views of the NIH or ABCD consortium investigators. This work was supported by the European Research Council under the European Union's Horizon 2020 research and Innovation program (802998), the Research Council of Norway (#288083, #300767, #323951), and the South-Eastern Norway Regional Health Authority (#2021070, #2023012, #500189).

## Author contributions

Linn B. Norbom led the project as first author and was central to all stages. Espen M. Eilertsen consulted on genetics, set up the initial heritability and ACE analyses and code, drafted the genetic-modelling methods, and reviewed and revised the manuscript. Andreas Dahl consulted on genetic modelling in the ABCD sample, guided variable selection based on ABCD literacy, and reviewed and revised the manuscript. Valerie Karl contributed ABCD expertise, advised on SES and variables related to population stratification, and reviewed and revised the manuscript. Lars T. Westlye

provided access to the ABCD dataset, advised on study framing and key considerations, and reviewed and revised the manuscript. Christian K. Tamnes, project PI, co-conceived the study, defined the scope and analytic strategy, oversaw the work, and reviewed and revised the manuscript.

## Competing interests

The authors declare no competing interests.
