## [Transparent Peer Review file · Communications Biology]

Socioeconomic context influences the heritability of child cortical structure

Corresponding Author: Dr Linn Norbom

This manuscript has been previously submitted at another journal. This document only contains information relating to versions considered at Communications Biology.

Version 0:

Reviewer comments:

Reviewer #1

(Remarks to the Author)

This study explores how SES mediates the heritability of cortical structures within a population of American children. The paper suggests that variance of grey/white matter contrast (GWC) is genetically influenced among children in high SES families. Whereas GWC of children in low SES families are more influenced by environmental factors rather than genetic heritability. This influence was not found within cortical thickness, sulcal depth, and surface area variance. Instead, surface area was highly genetically influenced regardless of SES. Furthermore, genetic heritability of general cognitive abilities is also influenced by SES, with children from high SES families exhibiting higher heritability. However, genetic and environmental influences on general cognitive abilities were similar in children from low SES families. The paper discusses how these results are in line with the "Saunders hypothesis" that posits high SES as a protective factor, reducing genetic potential expression amongst this population.

The data and analyses support the main research question and aim of the paper. The figures were mostly readable and understandable. The discussion addressed the results and possible reasons behind it. There is some missing information regarding background factors and some unclear terms that should be explained before this study is considered for publication.

Specific comments:

Major:

1. Apparently C (shared environment) and E (unique environment) just represent the unexplained variance in the model, which isn't explained particularly clearly in the article. However, as there is no information on twins / other siblings in the final sample, it is hard to interpret these results. Could the small effect of shared environment be related to there being a small number of people in same families? Furthermore, C was missing from statistical analysis section in preregistration. Elaborate on the selection to include C and give relevant information to interpret it (such as number of participants with family members in the study (MZ twins, DZ twins, other siblings)).
2. Please explain (in the manuscript, too), why the PCA approach was chosen for the creation of the SES index. Assuming the authors stay with that method, some exploration such as scatter plots showing the correlation with the individual SES metrics would add value (as they would confirm that the SES index can be interpreted as straightforwardly as the authors claim). Another approach might be to run the same analysis with a SES variable build using an alternative method (such as summing the standardized individual category scores) to see if the results remain mostly the same.
3. Is the pubertal status of the participants known? The onset of puberty affects brain development (e.g., <https://doi.org/10.1016/j.dcn.2023.101261>), and lower SES has been associated with earlier puberty (e.g., <https://www.sciencedirect.com/science/article/abs/pii/S1054139X21000707>). Those participants whose puberty has started could have a big effect on the results and confirmatory analyses should be run with prepubertal children only. If data is not available, this should be discussed as a limitation.

Minor:

4. The scope of the journal seems to target a wider range of professionals. Those within the psychology and child development disciplines would find this paper useful, and so brief explanations of genetic components would provide further context (e.g., h^2 , additive genetics) and improve readability for a wider audience.

5. Make the formatting of Figure legends uniform. Also there are two Figure 2's, the latter one's legend has more detail than others.
6. Authors could add discussion on whether occupation would add much value to the SES assessment (presumably rather highly correlated with education and/or income).
7. Use of FreeSurfer 6.0.0 should be mentioned as a limitation, as it is becoming rather outdated (version 8.0.0 is already out).
8. For figure 6, the type of "correlation" (Pearson, Spearman, partial) is not specified. Also p-values should be indicated somehow, e.g., * indicates $p < 0.05$, ** < 0.01 etc.
9. In supplementary Figure 1, the proportions are so small that it's impossible to tell the difference between most of the top rows. Can it be made a bit bigger?
10. How well a specific income matches the needs of the family depends on the number of people in the household and the geographical location. This should be considered in the analyses or mentioned as a limitation of the used SES index.

Reviewer #2

(Remarks to the Author)

This is a strong and timely paper that explores how socioeconomic status (SES) influences the balance of genetic and environmental contributions to brain structure in children. Using a large, well-curated dataset from the ABCD study (over 9,000 participants), the authors apply solid methods to estimate SNP-based heritability across multiple cortical features, including thickness, surface area, sulcal depth, and grey/white matter contrast (GWC). The key finding that children from lower-SES backgrounds show more environmental influence on brain structure, while genetic effects are stronger in high-SES settings is compelling and novel. It builds on concepts like the Scarr-Rowe hypothesis and expands them from cognition to brain structure, which hasn't been done at this scale before. The study's strengths lie in its massive sample size, careful quality control, and transparent pre-registration. The SES index is thoughtfully constructed, combining income, education, and neighborhood deprivation, offering a well-rounded measure. Overall, this is an important and meaningful contribution, but several key issues, particularly regarding ancestry stratification and interpretation of environmental effects, need to be addressed before the manuscript is ready for publication.

Major issues:

1. The paper mostly sticks to associations, but occasionally it hints at causality, especially with phrases like "reflects lived experience." I'd suggest rephrasing some of those moments to keep it clear that these are correlational findings.
2. Unique environment (E) is doing a lot of work in the interpretation, but it also includes measurement error and other noise. A short explanation of what E does and doesn't capture would be helpful, especially since it's central to the key findings.
3. What about population stratification? From what I can tell, the authors didn't limit the analyses to participants of a specific ancestry, e.g. European. They included 32 genetic PCs, which helps, but it's not a full solution. SES and ancestry are correlated in US samples, and this could bias both heritability estimates and SES interactions. The authors should be more upfront about this and ideally run a sensitivity analysis on a genetically homogeneous subsample.
4. Why leave out occupation in SES? I understand the challenges in ranking occupations, but it's still a standard SES component. Could the authors say more about this choice, or test whether adding occupation changes anything?

Minor issues:

2. The plots in Figures 2–5 are informative, but their interpretability would benefit from brief, in-panel legends or arrows indicating trends (e.g., "higher E in low SES").
3. In the Methods - Statistical analyses, "represented the cortical or cognitive metric" appears to be missing a variable label or symbol.

Reviewer #3

(Remarks to the Author)

The manuscript "Socioeconomic context influences the heritability of child cortical structure" examines whether the heritability of cortical structure in adolescence is affected by socioeconomic status. This is an important and well-executed study that addresses a critical gap in the literature. Overall, I found the manuscript well-written and methodically sound and I applaud the authors for pre-registering the study. I have the following questions/comments, which could help strengthen the manuscript.

Major comments

I noticed that there is no mention of the measurement error component of unique environmental variance throughout the manuscript. Do you think the lower heritability of cortical folds and intensity could relate to differences in measurement reliability between the different cortical measures?

Do you believe that the SES distribution contained here adequately captures low-SES families, given the well-known difficulties in attracting such families to research studies?

While Falconer's formula provides a quick estimate of heritability, it is an outdated option compared to structural-equation modelling options (i.e., OpenMx). It would help to strengthen your comparisons if your twin-based heritability estimates were produced using a more modern method.

Do you believe your sample had adequate power to detect GxE interaction effects?

Was there an association between scanner site & SES (i.e., did some sites collect a greater proportion of lower SES participants)? If so, it would be nice to see some further evidence that these site effects were entirely removed through data harmonisation.

Minor comments

I'm uncertain what the difference is between h^2 and A in the figure captions. A is additive genetic variance (but not expressed as a proportion of total phenotypic variance?).

Reviewer #4

(Remarks to the Author)

I co-reviewed this manuscript with one of the reviewers who provided the listed reports. This is part of the Communications Biology initiative to facilitate training in peer review and to provide appropriate recognition for Early Career Researchers who co-review manuscripts.

Version 1:

Reviewer comments:

Reviewer #1

(Remarks to the Author)

The authors have addressed my concerns and I suggest accepting the article for publication.

Reviewer #2

(Remarks to the Author)

The authors have answered my questions and addressed my concerns very well.

Reviewer #3

(Remarks to the Author)

The authors have provided thorough and well-documented responses to my comments and queries. I have no further comments.

Reviewer #4

(Remarks to the Author)

I co-reviewed this manuscript with one of the reviewers who provided the listed reports. This is part of the Communications Biology initiative to facilitate training in peer review and to provide appropriate recognition for Early Career Researchers who co-review manuscripts.

Response letter,

We truly thank the editor and reviewers for taking the time to give positive and constructive feedback. In particular, the reviewers' suggestions to clarify heritability, its distinction from additive genetics, and the scope of the environmental components have substantially improved our manuscript. Below we have responded in detail to each of the reviewer comments, and revised text is marked in green.

Reviewer #1

This study explores how SES mediates the heritability of cortical structures within a population of American children. The paper suggests that variance of grey/white matter contrast (GWC) is genetically influenced among children in high SES families. Whereas GWC of children in low SES families are more influenced by environmental factors rather than genetic heritability. This influence was not found within cortical thickness, sulcal depth, and surface area variance. Instead, surface area was highly genetically influenced regardless of SES. Furthermore, genetic heritability of general cognitive abilities is also influenced by SES, with children from high SES families exhibiting higher heritability. However, genetic and environmental influences on general cognitive abilities were similar in children from low SES families. The paper discusses how these results are in line with the "Saunders hypothesis" that posits high SES as a protective factor, reducing genetic potential expression amongst this population.

The data and analyses support the main research question and aim of the paper. The figures were mostly readable and understandable. The discussion addressed the results and possible reasons behind it. There is some missing information regarding background factors and some unclear terms that should be explained before this study is considered for publication.

Major comment 1:

A. Apparently, C (shared environment) and E (unique environment) just represent the unexplained variance in the model, which isn't explained particularly clearly in the article.

It is great that the reviewer made us aware that we had been unclear when presenting our ACE variance components. It is common practice in quantitative genetics to model measured covariates (age, sex, principal components relating to ancestry and SES) as fixed effects, and the genetic and environmental components of the phenotype as random effects. In our study, we considered additive genetic effects (a), and two types of environmental effects: those shared by relatives (c) and those specific to the individual (e). Thus, *all three* components (A, C and E) decomposed the unexplained/residual variance after conditioning on the measured covariates. The different types of random effects were separated because they have different covariance structures specified across individuals. If genetic variants or environmental properties had been measured, they could also have been modelled with fixed effects. Following reviewer feedback, we have within the "Statistical Analyses" section (page 25) expanded the description of our ACE components, and explicitly noted that classical measurement error is included in the E component, as requested by the other three reviewers:

“For each imaging metric and the general cognitive ability score, we fit a mixed effects “baseline model” using the svcm package in R. These models, for individual i in family j , include all main effects but no interaction terms and were specified as:

$$y_{ij} = \mathbf{w}'_{ij}\boldsymbol{\gamma} + \beta s_j + a_{ij} + c_j + e_{ij}.$$

In this linear model, (y) represented the cortical or cognitive metric, $\boldsymbol{\gamma}$ was the fixed effects of the covariates (e.g. age, sex and 32 genetic PCs derived by Fan et al. (2023), related to minimizing population stratification), and β was the fixed effect of the SES index (s).

The term a was a person-varying random additive genetic effect, distributed across individuals according to the GRM with variance σ_a^2 . For example, for siblings k and l , the GRM at position kl has a value of $\approx 1/2$, reflecting their estimated sharing of genetic material. If the siblings are monozygotic twins, the value is ≈ 1 . These values represent the expected correlation in phenotypes that arise because of additive genetic effects. Of note, other sources of variation may lead to similar patterns, for example if monozygotic twins are treated more similar by their parents. The term c was a family-varying random effect, independent across families with variance σ_c^2 . However, other sources that increase resemblance among relatives may lead to similar patterns, for example non-additive genetic effects or assortative mating. The term e was a person-varying random environmental effect independent across individuals with variance σ_e^2 . This term has the same structure as the residual in other linear models and capture other sources of variability that is not correlated across individuals, such as classical measurement error.”

B. However, as there is no information on twins / other siblings in the final sample, it is hard to interpret these results. Could the small effect of shared environment be related to there being a small number of people in same families?

We appreciate the reviewer’s observation that our original manuscript lacked detail on family structure, which is essential for modelling genetic and shared environmental components. We have in the revised “Participants” section (page 18-19) provided such information:

“The final sample thus consisted of 9080 children (4266=females) aged 8.9-11.1 years (mean = 9.9 SD= 0.63). Two individuals who were assigned intersex-male at birth were recoded as male to enhance statistical robustness while ensuring the inclusion of all relevant individuals to our study. The sample comprised 6536 children from singleton families and 2544 children from sibship families, with counts reflecting enrolled siblings rather than the actual household size. Of these, 2428 children came from two-sibling families, 111 from three-sibling families, and five from a five-sibling family. The dataset included 670 twin pairs – 261 monozygotic and 409 dizygotic – plus 7 complete triplet sets and 1 incomplete set in which only two of the three siblings met inclusion criteria.”

As the reviewer points out, we agree that the small, estimated contributions of shared environments also could be an artifact of the relatively small number of enrolled sibship families, most of which were families with two enrolled siblings. Williams (1993) pointed out the inherent difficulty of statistically separating shared genetic and environmental effects in the context of the classical twin design. In

contrast, our design includes larger families than sibling-pairs and measure genetic similarity from molecular data, which should reduce the problem (which was also pointed out by Williams). We have in the revised “Discussion” section (page 15) added this issue:

“Shared environment, or specifically being part of the same family, was minimally associated with cortical structure in our study. Both the inclusion of families with more than two participating siblings and the use of molecular genetic data help to better separate shared genetic from shared environmental effects (Williams, 1993). The small shared environment contribution could still reflect limited statistical power, as only 28% of participants had enrolled siblings in the dataset, most of whom were from two-sibling families.”

C. Furthermore, C was missing from statistical analysis section in preregistration. Elaborate on the selection to include C and give relevant information to interpret it (such as number of participants with family members in the study (MZ twins, DZ twins, other siblings)).

We thank the reviewer for catching this undocumented discrepancy. As the reviewer correctly points out our preregistration described an AE model, thus parsing variance into one genetic and one general non-genetic component. As our original hypothesis centered on genetics, namely whether the Scarr-Rowe effect (i.e., that the genetic contribution to cognitive ability varies with SES) extends to brain structure, we in hindsight initially gave insufficient attention to non-genetic modelling. Once the study began and we realized that we could use the extensive family information to further parse non-genetics into variance shared within families (C), which we found informative, we adopted a classic ACE model.

In the revised “Statistical Analyses” section (page 26), we have documented this discrepancy between the original preregistration and the performed analyses:

“Under this model, the genetic and environmental components of variance do not depend on the SES index. Of note, the preregistration detailed a genetic (a) + non-genetic (e) model, but with extensive family membership data available, we chose to further parse the non-genetic signal by also modelling the variance shared between family members (c).”

Major comment 2: Please explain (in the manuscript, too), why the PCA approach was chosen for the creation of the SES index. Assuming the authors stay with that method, some exploration such as scatter plots showing the correlation with the individual SES metrics would add value (as they would confirm that the SES index can be interpreted as straightforwardly as the authors claim). Another approach might be to run the same analysis with a SES variable build using an alternative method (such as summing the standardized individual category scores) to see if the results remain mostly the same.

Our aim was to capture the overarching construct of SES as accurately as possible, and we chose principal-component analysis (PCA) to combine the individual SES sub-factors (see SI Fig. 3 for a biplot of sub-factor contributions). We chose a data

driven reduction as we had no a-priori indication that all sub-factors must contribute equally or with specific preset weights. Moreover, PCA is a widely accepted data driven reduction technique that assigns weights that maximize shared signal while down-weighting noisier indicators (possibly self-reported income for instance). Nonetheless, we agree with the reviewer that in many cases, advanced reduction methods add little benefit over simpler reductions (Clark et al., 2021) such as an equal-weight sum. As suggested by the reviewer, we therefore created a SES-index by summing the standardized category scores and correlated each sub-factor, the original SES-index (SES_PCA), and the new summed SES-index (SES_Sum) with one another in a correlation matrix (Review Figure 1).

Reviewer Figure.1 Correlation matrix of SES-subfactors, the original SES-index (SES_PCA) and the alternative summed SES-index (SES_Sum).

The PCA-derived and summed SES indexes were virtually identical ($r = 0.9987$), thus capturing the sub-factors equally well, with each sub-factor correlating between 0.71 and 0.87 with the two indices. Accordingly, we retained our original SES-index, but have in the revised “Measurements of Socioeconomic Status and Neighbourhood Deprivation” section (page 20) clarified our rationale for using PCA, and now specify sub-factor correlations with the SES-index:

“To create a single overarching SES index, we reduced z-scored parental education, parental income, and neighborhood deprivation by principal component analysis

(PCA) in R using the “prcomp” package. PCA was chosen as it is a conventional data driven reduction that avoids a-priori assumptions that SES-subfactors contribute equally or with preset weights and maximizes shared variance while remaining sensitive to noise differences. The first component, which explained 65% of the total variance, was extracted as the overarching “SES-index”. It correlated strongly with parental education ($r = 0.84$), parental income ($r = 0.87$), and neighborhood deprivation ($r = -0.71$). Higher values reflect higher income and education, and lower neighborhood deprivation (SI Figure 3).”

Major comment 3: Is the pubertal status of the participants known? The onset of puberty affects brain development

(e.g., <https://doi.org/10.1016/j.dcn.2023.101261>), and lower SES has been associated with earlier puberty

(e.g., <https://www.sciencedirect.com/science/article/abs/pii/S1054139X21000707>). Those participants whose puberty has started could have a big effect on the results and confirmatory analyses should be run with prepubertal children only. If data is not available, this should be discussed as a limitation.

We appreciate the reviewer’s concern that, because pubertal maturation influences cortical structure and, on average, begins earlier in lower-SES children, pubertal status could partly account for the observed SES moderation of genetic influence. Our goal was to test whether SES moderates genetic contributions on cortical structure, not to isolate individual pathways. Earlier puberty is itself a characteristic of lower SES. Although we have data on pubertal status (Beck et al., 2023), as well as other central variables that influence both SES and brain structure, such as early-life adversity (Beck et al., 2025), and pregnancy and birth related complications (Lindseth et al., 2025) regressing these factors out would remove variance that legitimately reflects the circumstances of low-SES children and would progressively undermine the construct we seek to study.

For the reviewer’s reference, we in the following, provide information on pubertal status in our final sample. Our final sample ranged in age from 8.9 to 11.1 years ($M = 9.9$, $SD = 0.63$) and we had complete care-giver-reported pubertal status for roughly half of these participants ($n = 4,517$). In ABCD, puberty is assessed with the five-item Pubertal Development Scale (PDS), which quantifies growth spurts, body-hair growth, and skin changes in all children, plus breast development and menarche in females and voice change and testicular growth in males. Each item is rated on a four-point scale (1 = no development, 2 = barely begun, 3 = definitely under way, 4 = development complete; menarche is coded 1 = not begun, 4 = begun). Based on the reviewer comment we computed sex-specific sum scores for participants with complete PDS data, where a minimum score of 5 marks a pre-pubescent child, and a maximum score of 20 reflects complete pubertal development.

Reviewer Figure 2. Pubertal development status. The figure shows the distribution of pubertal development score for males and females included in the final sample with complete pubertal data (n = 4517).

Reviewer Figure 3. Pubertal development across the SES-spectrum. The figure shows individual pubertal development score plotted as a function of the SES-index. The fit lines reflect LOESS-smoothed trends (locally estimated scatterplot smoothing).

Based on visualizations, our cohort is largely pre- to early-pubescent (Reviewer Figure 2). The LOESS-smoothed trend in Reviewer Figure 3 indicates that boys and

girls in the higher-SES group cluster around a total PDS score of ~8, whereas their lower-SES peers are slightly more advanced (boys ~9; girls approaching 10). Wilcoxon rank-sum tests after splitting the SES-index at the median, confirmed this pattern: lower-SES children had higher PDS scores than their higher-SES peers in both boys ($W = 822\ 786$, $p = 0.001$) and, more strongly, girls ($W = 347\ 118$, $p < 0.001$), indicating a modest but statistically significant low-SES-related advance in pubertal status.

Reviewer Figure 4. Pearson's correlations of the SES-index and cortical structure. The figure shows correlations of the SES-index and global cortical structure before (left) and after (right) controlling for total PDS.

Finally, when we correlate the SES index with global brain metrics directly and then repeat the analysis while controlling for total PDS scores (partial Pearson correlations), the SES-cortical structure association looks highly similar (bottom rows of Reviewer Figure 4). Even so, the moderation of SES on the *genetic component* of cortical structure could show a different pattern.

In the revised "Statistical Analysis" section (page 26), we now provide an explicit justification for not modelling certain variables that influence both SES and brain structure.

"Adjustments for multiple comparisons were performed using the Benjamini-Hochberg procedure for false discovery rate (FDR), separately for global and lobar analyses. Statistical significance was defined as FDR-adjusted $p \leq 0.05$. Of note, we left factors associated with SES and brain structure (e.g. pubertal timing, early-life trauma, and pre-/perinatal risk) unadjusted by design, as they relate to the SES context of interest. Future mechanism-focused studies should model them explicitly."

Minor comment 4: The scope of the journal seems to target a wider range of professionals. Those within the psychology and child development disciplines would find this paper useful, and so brief explanations of genetic components

would provide further context (e.g., h^2 , additive genetics) and improve readability for a wider audience.

This is a great idea by the reviewer, and a similar comment was also made by reviewer 3. We have in the revised last paragraph of the “Introduction” section (page 4) attempted to clarify what our heritability and additive genetic component signifies:

“In this preregistered study, we analysed cross-sectional data from over 9,000 children and their parents, including 2544 children with at least one co-enrolled sibling, from the Adolescent Brain Cognitive Development (ABCD) Study. Genetic effects were assessed via SNP genotypes, child cortical structure via MRI based global and lobar cortical thickness, surface area, sulcal depth, and grey/white-matter contrast (GWC), and SES via a composite of parental income, education, and neighbourhood deprivation. We first estimated the overall heritability ($h^2 = A / (A + C + E)$) of cortical structure, where A is the additive genetic variance, C is the environmental variance shared by siblings, and E is the environmental variance unique to the child. Because h^2 is a proportion of total variance attributable to genetics, it depends on the denominator; for example, increasing environmental variance (C or E) lowers h^2 even if A is unchanged. Next, we tested whether the absolute variances A, C, and E differed across socioeconomic contexts. Additionally, we examined the genetic and environmental effects on cognitive abilities within the same sample.”

Minor comment 5. Make the formatting of Figure legends uniform. Also there are two Figure 2’s, the latter one’s legend has more detail than others.

We thank the reviewer for noticing discrepancies in the legend formatting as well as the misnumbering of Figure 3 as a second Figure 2. In the revised manuscript, we have corrected both issues, and have added an example of the final legend that will be used for Figure 2-6, with the imaging or cognitive metric substituted as appropriate:

“Figure 4. Genetic and environmental variance of mean grey-white matter contrast (GWC) across SES. The image displays splines representing heritability ($h^2 = A / (A + C + E)$); brown, along with its components: additive genetics (A; green), shared environment (C; dark orange) and unique environment (E; yellow) for mean GWC across the standardized SES construct. The density distribution of SES is shown as a shaded area below the x-axis. The dotted grey line shows the total phenotypic variance ($A + C + E$).”

Minor comment 6. Authors could add discussion on whether occupation would add much value to the SES assessment (presumably rather highly correlated with education and/or income).

We agree with the reviewer that, since we were unable to include one of the “big 3” SES indicators (Long & Renbarger, 2023), further information on occupation is warranted. As first theorized by sociologist Pierre Bourdieu, occupation is often interpreted as capturing *symbolic capital* or “prestige” (Bourdieu, 2011). It could also proxy aspects of the physical and psychosocial work environments which again can influence health (Fujishiro et al., 2010). Income, education, and occupation are generally moderately correlated (Farah, 2017), and our prior work in a separate

neurodevelopmental US sample (n = 504) likewise showed correlations of $r \approx 0.6$ between occupation and each of the other indicators (Norbom et al., 2022).

It is highly speculative to determine how including parental occupation might have influenced our findings. Due to its moderate correlation with the other SES indicators, combined with our use of only the first principal component as the SES-index, one could speculate that most of occupation's shared variance would already have been captured in that component. We would therefore expect broadly similar findings for our SES-index. Any occupation-specific variance not shared with the dominant income or education signal would likely load onto later PCA components, which were not analyzed in the present study.

We have in the revised "limitations" section (page 17) broadened our description of occupational status:

"There are several limitations to our study. First, although we aimed to create a broad SES construct, we were unable to reliably model the key subfactor **parental occupation**. **Occupation shows moderate correlations with income and education (Farah, 2017; Norbom et al., 2022), and captures symbolic capital or prestige (Bourdieu, 2011) and work-environmental conditions linked to physical and mental health (Fujishiro et al., 2010).**"

Minor comment 7. Use of FreeSurfer 6.0.0 should be mentioned as a limitation, as it is becoming rather outdated (version 8.0.0 is already out).

Based on this reviewer comment we realized that we had originally written the incorrect FreeSurfer version. We have in the revised "MRI acquisition, quality control, preprocessing, and scanner harmonization" section (page 22) updated this information based on release notes for ABCD 5.1:

"We relied on FreeSurfer version **7.1.1** runs provided by the ABCD consortium to perform volumetric segmentations and cortical surface reconstructions."

Minor comment 8. For figure 6, the type of "correlation" (Pearson, Spearman, partial) is not specified. Also p-values should be indicated somehow, e.g., * indicates $p < 0.05$, ** < 0.01 etc.

We thank the reviewer for noticing this and have specified in the revised figure legend (page 24) that the correlations reported are Pearson's correlations. We chose not to include p -values, as these correlations were intended for descriptive purposes rather than formal hypothesis testing.

"Figure 7. **Pearson's** correlations between relevant variables..."

Minor comment 9. In supplementary Figure 1, the proportions are so small that it's impossible to tell the difference between most of the top rows. Can it be made a bit bigger?

We thank the reviewer for this suggestion and have in the revised Supplementary Information (SI) increased the size of Supplementary Figure 1.

Minor comment 10. How well a specific income matches the needs of the family depends on the number of people in the household and the geographical location. This should be considered in the analyses or mentioned as a limitation of the used SES index.

We agree with the reviewer and have clarified this in the revised “limitations” section (page 17):

“There are several limitations to our study... **Second, the index is not sensitive to family size (“income-to-needs”) or geographic variation in cost of living across the US, and does not capture subjective, relative or cultural aspects of SES (Liu et al., 2004; Yosso, 2005).**”

Reviewer #2

This is a strong and timely paper that explores how socioeconomic status (SES) influences the balance of genetic and environmental contributions to brain structure in children. Using a large, well-curated dataset from the ABCD study (over 9,000 participants), the authors apply solid methods to estimate SNP-based heritability across multiple cortical features, including thickness, surface area, sulcal depth, and grey/white matter contrast (GWC). The key finding that children from lower-SES backgrounds show more environmental influence on brain structure, while genetic effects are stronger in high-SES settings is compelling and novel. It builds on concepts like the Scarr-Rowe hypothesis and expands them from cognition to brain structure, which hasn't been done at this scale before. The study's strengths lie in its massive sample size, careful quality control, and transparent pre-registration. The SES index is thoughtfully constructed, combining income, education, and neighborhood deprivation, offering a well-rounded measure. Overall, this is an important and meaningful contribution, but several key issues, particularly regarding ancestry stratification and interpretation of environmental effects, need to be addressed before the manuscript is ready for publication.

Major comment 1. The paper mostly sticks to associations, but occasionally it hints at causality, especially with phrases like “reflects lived experience.” I'd suggest rephrasing some of those moments to keep it clear that these are correlational findings.

We completely agree with the reviewer that it is essential to use language consistent with the analyses performed, particularly when discussing genetics, which can be perceived as holding a causal role by default. We were also mindful of possibly making the message within the abstract and discussion unclear if only using statistical (“explaining variance”) or purely correlational (“associated with”) terms. Thus, in some instances, we used terms common in behavioral genetics, such as “influence” and “shape”, which we acknowledge are causal by strict definition. Based on the reviewer comment, we re-assessed all descriptions of findings and revised when appropriate as documented at the end of this response.

Throughout the text we have used the term “influence” in reference to both SES “influencing” ACE components and ACE components “influencing” cortical structure. Based on Merriam-Webster’s definition of influence as an indirect or intangible effect of causation, we have revised all non-appropriate instances. We retained a single instance where we were speculating on mechanistic pathways rather than describing statistical results.

As noted by the reviewer we used the term “reflect” in the final sentence of the abstract and conclusion. This term is non-causal in this specific context and is similar to the word “mirror”, or according to Merriam-Webster “to make (something) manifest or apparent” and we therefore retained our original wording.

In the results section we intentionally used causal language (i.e. “due to”) as we were indicating that the higher h^2 values in our models were caused by higher A and lower E and therefore keep original wording. We also found a single use of “shaping” when discussing GWC-related findings. We agree that this term is synonymous with “influence” and therefore replaced it with “proportion of variance”. We have also adjusted all of SI material accordingly.

Causal-language revisions:

-Abstract (page 2):

- “We then tested whether genetic and environmental **contributions** were moderated by parental SES.”
- “However, among children from lower-SES backgrounds, cortical differences were less genetically **related** and **more uniquely environmentally related**, at times exceeding genetic contributions.”

-Results:

- Sub-header: “Genetic **contribution** to cortical structure and moderation by parental SES” (page 5):
- “In contrast, for children from low-SES backgrounds, unique environmental **contributions** exceed genetic contributions, **accounting for a greater proportion of the variance in GWC** under these circumstances.” (page 9)

-Discussion:

- “In a sample of over 9000 10-year-old children from the US, we found that differences in cortical thickness and surface area were primarily **accounted for** by genetic variation, while cortical folding and intensity showed moderate heritability.” (page 13)
- “...where regional cortical structure appears to be more **related to** unique environmental factors, at times exceeding genetic **contributions**.” (page 14)
- “...unique environmental factors appeared to play a larger role, corresponding to-, or even exceeding the **contribution** of genetics.” (page 15)
- “Shared environment, or specifically being part of the same family, **was minimally associated with** cortical structure in our study.” (page 15)
- “...used 705 Australian twins and their siblings to assess whether the heritability and additive genetic **contributions** on white matter fractional anisotropy.” (page 15)
- “Among socioeconomically advantaged children, inherited predispositions appear to **play a larger role in** brain structural **differences**.” (page 17)

Major comment 2. Unique environment (E) is doing a lot of work in the interpretation, but it also includes measurement error and other noise. A short explanation of what E does and doesn't capture would be helpful, especially since it's central to the key findings.

We completely agree and refer to our detailed response to Reviewer 1 major comment 1, which raised a similar point. In short, we have revised the manuscript to provide a clearer description of our ACE components.

Major comment 3. What about population stratification? From what I can tell, the authors didn't limit the analyses to participants of a specific ancestry, e.g. European. They included 32 genetic PCs, which helps, but it's not a full solution. SES and ancestry are correlated in US samples, and this could bias both heritability estimates and SES interactions. The authors should be more upfront about this and ideally run a sensitivity analysis on a genetically homogeneous subsample.

We thank the reviewer for this comment and agree that ancestry-based environmental and genetic contributions often co-vary in ways that are difficult to disentangle. For this reason, heritability estimates in ancestrally diverse samples are often likely somewhat inflated, although it depends on the environmental sensitivity of the outcome.

In our SES moderation analysis, population stratification arises if the magnitude of ancestry-correlated *environmental* effects erroneously captured by A varies across SES. This could occur if ancestry is unevenly distributed, such that one end of the spectrum is more homogeneous, or both ends are homogeneous but of differing ancestry. As we reported in a previous study, also using the ABCD sample, higher-SES children were disproportionately of European ancestry, while children from lower-SES settings were disproportionately from minority ancestry groups, including African ancestry (Norbom et al., 2023).

Although no study can perfectly account for all aspects of population structure, we pre-registered that we would mitigate ancestral variance by adding 32 genetic principal components as fixed effects. As expected, several components show modest and minor correlations with the SES-index (New SI Figure 5).

New SI Figure 5. Pearsons correlations of each of the 32 genetic principal component and the SES-index.

Our rationale for using genetic components as fixed effects was twofold: First, there is no clear consensus on what constitutes a homogeneous ancestral group, i.e. how *genetically* similar individuals must be for population stratification to be considered negligible. Even if such criteria were established, population-based samples rarely contain the information needed to reliably extract only these individuals and instead typically rely on subjective or self-reported ancestry. Second, restricting analyses to genetically similar individuals would disproportionately exclude individuals at the lower end of the SES spectrum, since the largest homogeneous group is typically of European ancestry, making SES moderation assessments unstable. Still, based on the reviewer’s comment, we attempted to construct a more genetically homogeneous subsample. Although some degree of population stratification is likely to remain within this subgroup, it should be less pronounced than in the full sample.

We examined the first 10 genetic PCs because they capture the major axes of genetic variation. From these, we retained PCs whose correlation with SES was ≥ 0.10 in magnitude (PCs 1–5 and 7). We then applied k-means clustering ($k=2$) to these PCs and selected the larger cluster as the similar-ancestry subgroup ($n=7,445$). The excluded cluster appeared to be disproportionately lower-SES (Blue group; New SI Figure 6) and predominantly parent-reported as Black (1,189), which, of note, can diverge from genetic ancestry, the relevant factor when aiming to reduce population stratification. It contained only three White- and no Asian individuals. In contrast, the included subgroup was parent-reported predominantly as White (5,063) and Hispanic (1,597), contained all Asian participants (120), and had only 16 Black participants (new SI Table 5).

New SI Figure 6. Box plot of the distribution of the SES-index across the homogeneous subsample (group 1) and the excluded individuals (group 2).

Cluster	Asian	Black	Hispanic	Other	White
1	120	16	1597	649	5063
2	0	1189	120	323	3

SI Table 5. Parent-reported ethnicity for group 1, included in the sensitivity analyses as a more genetically homogeneous subgroup, and for group 2, which was excluded from these analyses.

To maximize similarity with the original analyses, we used the scaling derived from the original sample and then re-ran the heritability estimates and SES moderation analyses of the global imaging metrics, restricting the analyses to the homogeneous subsample.

Our results showed that overall heritability estimates were highly similar to the original results (New SI Table 2). For cortical morphometry, estimates were never lower and at times slightly higher, which argues against population stratification being a source of bias. For GWC, estimates differed by 0.04 points between the homogeneous and original more genetically diverse sample, indicating at most a negligible effect of population stratification on heritability estimates.

Global imaging metric	Original sample (n = 9080)	Ancestrally homogeneous sample (n = 7445)
Thickness	$H^2 = 0.82$	$H^2 = 0.83$

Area	$H^2 = 0.82$	$H^2 = 0.88$
Sulcal depth	$H^2 = 0.57$	$H^2 = 0.59$
GWC	$H^2 = 0.57$	$H^2 = 0.53$

New SI Table 2. The table shows the heritability estimates for the global imaging metrics in the original sample and in the ancestrally homogeneous subsample.

Our SES moderation analyses yielded overall similar chi-square values (New SI Table 3). Notably, global cortical thickness, previously reported as non-significant (corrected $p = 0.051$), was significant in the more genetically homogeneous sample. Sulcal depth again showed null findings, while the moderation effect of SES on the genetic and environmental components of GWC replicated and appeared slightly stronger. In contrast, the SES moderation of surface area, which had been reported as non-significant, but which had showed borderline effects proportionate to cortical thickness, was absent in the more genetically homogeneous sample. While the reduced sample size limits firm conclusions, it is also possible that population stratification contributed to the borderline effects, which, to be clear, were reported as not significant in the original paper as well.

Metric	Original sample (n = 9080)	Homogeneous sample (n = 7445)
Thickness	$\chi^2 = 8.27, p = 0.041, pFDR = 0.051$	$\chi^2 = 10.94, p = 0.012, pFDR = 0.024$
Area	$\chi^2 = 8.34, p = 0.040, pFDR = 0.051$	$\chi^2 = 0.01, p = 1.000, pFDR = 1.000$
Sulc. depth	$\chi^2 = 1.08, p = 0.781, pFDR = 0.781$	$\chi^2 = 1.25, p = 0.740, pFDR = 0.987$
GWC	$\chi^2 = 13.59, p = 0.004, pFDR = 0.010$	$\chi^2 = 16.24, p = 0.001, pFDR = 0.004$

New SI Table 3. The table shows the chi-squared values and the accompanying p -values for the global imaging metrics in the original sample and in the ancestrally homogeneous subsample.

We have in the revised manuscript and SI material added these sensitivity analyses:

Results section:

- Cortical thickness. “Heritability estimates for all imaging metrics, whether derived from a more genetically homogeneous subsample (SI Figure 5, SI Table 5) or obtained using a twin-based method, were highly similar and are presented in the SI Results (SI Table 2-3).”
- Surface area. “Likelihood ratio tests yielded a non-significant interaction between the SES index and genetic and environmental factors after correction ($\chi^2=8.34, Df=3, p=0.04$, corrected $p = 0.051$), thus providing no indication that the heritability of global surface area differs across SES levels. While sensitivity analyses with a more genetically homogeneous subsample produced highly similar results across global imaging metrics (SI Table 3), the SES-index moderation was weaker for surface area.

SI Methods:

Construction of a genetically homogeneous subsample

To mitigate residual population stratification, we constructed a more genetically homogeneous subsample using the genetic principal components (PCs). We

examined the first 10 PCs and retained those whose correlation with the SES index was at least 0.10 in magnitude (PCs 1–5 and 7). We then applied k-means clustering (k=2; nstart=25) and retained the larger cluster as the genetically homogeneous subsample (n=7,445). The excluded cluster appeared to be disproportionately of lower-SES (Blue group; SI Figure 5) and predominantly parent-reported as Black (1,189). In contrast, the included subgroup was parent-reported predominantly as White (5,063) and Hispanic (1,597), contained all Asian participants (120), and had only 16 Black participants (SI Table 5).

SI Results:

Heritability of cortical structure within genetically homogeneous sample

Overall heritability estimates of cortical structure within the more genetically homogeneous subsample were highly similar to the original analyses (SI Table 2). SES moderation analyses also produced results consistent with the main findings, apart from surface area, where the effect weakened (SI Table 3). This may either reflect reduced power in the sensitivity analysis or inflated (albeit nonsignificant) moderation in the original sample due to population stratification.

Major comment 4. Why leave out occupation in SES? I understand the challenges in ranking occupations, but it's still a standard SES component. Could the authors say more about this choice, or test whether adding occupation changes anything?

As pre-registered, our rationale for not including occupation despite it being a classic quantitative indicator of SES was as follows: “While the questionnaire is detailed and includes occupations on a 22-point scale, it does not attempt to order them based on status. To maintain the integrity of our SES measure and avoid introducing subjective bias, we refrain from assigning our own rankings to reported occupations”. Moreover, we provided a concrete example of the first five categories, where from a subjective perspective, four of the five categories could have highly similar status. We suspect, although unconfirmed, that the status-neutral nature of the occupational questionnaire explains why other ABCD-based studies aiming to capture multiple aspects of SES also do not include occupation (Rakesh et al., 2021, 2022). Of note, we have previously worked with other US samples where occupational status was predictably ranked within the questionnaire for instance from unskilled employee to higher executive/major professional (Norbom et al., 2022).

In regard to the topic of occupation, we also refer to our response to Reviewer 1, Minor Comment 6, where we state that because occupation moderately correlates with income and education, and as we only used the first principal component as our SES index, thus capturing the dominant variance shared across indicators, it is unlikely that including occupation would substantially change our results.

Minor comment 5. The plots in Figures 2–5 are informative, but their interpretability would benefit from brief, in-panel legends or arrows indicating trends (e.g., “higher E in low SES”).

We appreciate the reviewer’s suggestion. We fear that adding in-panel legends or arrows would make the already dense figures overly cluttered. Instead, we have

added more extensive and consistent figure legend text, and refer to Reviewer 1, Minor comment 5.

Minor comment 6. In the Methods - Statistical analyses, “represented the cortical or cognitive metric” appears to be missing a variable label or symbol.

We believe our original sentence was correct, as “(y)” represented the cortical or cognitive metric. However, in the revised manuscript we have clarified this by adding a comma (page 25): “In this linear model, (y) represented the cortical or cognitive metric.”

Reviewer #3

The manuscript “Socioeconomic context influences the heritability of child cortical structure” examines whether the heritability of cortical structure in adolescence is affected by socioeconomic status. This is an important and well-executed study that addresses a critical gap in the literature. Overall, I found the manuscript well-written and methodically sound and I applaud the authors for pre-registering the study. I have the following questions/comments, which could help strengthen the manuscript.

Major comment 1:

A. I noticed that there is no mention of the measurement error component of unique environmental variance throughout the manuscript.

We completely agree that this should be included and refer to our detailed response to Reviewer 1 major comment 1, which raised a very similar point. In short, we have revised the manuscript to provide a clearer description of our ACE components.

B. Do you think the lower heritability of cortical folds and intensity could relate to differences in measurement reliability between the different cortical measures?

The reviewer raises an important point that reliability differences among sMRI metrics can influence the ACE and heritability estimates. As a pure intensity-based measure, GWC is more susceptible to non-biological variance, such as bias fields or coil sensitivity, affecting intensity values. This makes GWC noisier than geometry-based morphometry, and the additional noise is absorbed into the measurement error within E. Thus, even if the additive genetic variance (A) were comparable to that of other imaging metrics, an inflated E would reduce the heritability estimate. In contrast, folding, which also showed moderate heritability, has been reported to show high test–retest reliability (Pizzagalli et al., 2020) and in our own previous work we found it to be the least noisy of the four cortical metrics (Norbom et al., 2023, SI Discussion). In our main SES moderation analyses we report not only the heritability estimates but also the raw variance components, precisely to make such distinctions transparent.

Based on the reviewer’s comment, we first examined how our sMRI image quality variable (“defect topology,” reflecting the number of holes, bumps, and/or handles in the original surface prior to FreeSurfer correction) correlated with the global imaging metrics. As expected, sulcal depth showed the weakest association with defect

topology ($r = 0.03$), followed by surface area ($r = 0.16$), cortical thickness ($r = -0.24$), and GWC ($r = -0.40$), appearing most sensitive to noise. We then extracted the ACE components for each global metric (Reviewer table 1.)

Metric	A	C	E
Thickness	0.77	0.00	0.17
Surface area	0.56	0.07	0.05
GWC	0.56	0.00	0.42
Sulcal depth	0.55	0.00	0.41

Reviewer Table 1. The table shows the A, C, and E components for the global imaging metrics.

Additive genetic contributions were similar for surface area, GWC, and sulcal depth ($A \approx 0.55-0.56$) but higher for thickness ($A = 0.77$). Despite this, surface area and thickness showed comparable heritability, as the larger E component for thickness offsets its higher genetic variance. By contrast, the lower heritability of GWC and sulcal depth appears driven by substantially larger E. While this likely at least in part reflects reduced reliability of the GWC metric, it is less straightforward to attribute the elevated E in sulcal depth to noise alone. Moreover intensity metrics including GWC specifically have been shown to be more sensitive to SES-related environmental impact than standard morphometry (Norbom et al., 2022, 2023).

While a detailed discussion of what drives heritability estimates for global imaging metrics was not the focus of our study, we have added a sentence noting that GWC, as an intensity-based metric, is particularly prone to noise (page 14-15). Relatedly, following Reviewer 1 (major comment 1 and minor comment 4) we have additionally clarified that h^2 depends on the denominator (e.g., increasing environmental variance lowers h^2 even if A is unchanged) and that E also captures classical measurement error.

“We found that sulcal depth and GWC were both moderately heritable, with estimates approaching 60%. As a pure intensity metric, GWC is more susceptible to noise than geometry-based morphometry, potentially lowering the heritability estimate. Still, while no prior study has specifically assessed the heritability of these metrics in childhood, our results are generally in line with previous studies of related metrics or in young adulthood..”

Major comment 2. Do you believe that the SES distribution contained here adequately captures low-SES families, given the well-known difficulties in attracting such families to research studies?

This is an important point raised by the reviewer. Although ABCD recruitment was conducted through schools and informed by ethnicity, SES, and urbanicity, it is improbable that the SES distribution is representative of the U.S. population. This is due to entrenched recruitment biases (families with lower SES are less likely to participate in research) as well as exclusion biases (participants excluded during studies are often disproportionately from lower SES backgrounds). It is because of

exclusion biases we show the SES distribution not only for the included sample (SI Figure 4) but also for the excluded participants (SI Figure 2).

Still, there are no straightforward solutions to these biases, which affect all research attempting to tackle socioeconomic gradients. Moreover, depending on imaging from multi-site open-access initiatives such as the ABCD project, also means being removed from the recruitment process. We have specified that there are SES-related biases, in the revised limitations section (page 17):

“There are several limitations to our study...Moreover the index is not sensitive to family size (“income-to-needs”) or geographic variation in cost of living across the US, and does not capture subjective, relative or cultural aspects of SES (Liu et al., 2004; Yosso, 2005). Also, there are entrenched recruitment and exclusion biases when studying SES, as very low-SES families are less likely to participate in research and more likely to be excluded during data quality control.”

Major comment 3. While Falconer’s formula provides a quick estimate of heritability, it is an outdated option compared to structural-equation modelling options (i.e., OpenMx). It would help to strengthen your comparisons if your twin-based heritability estimates were produced using a more modern method.

We agree that Falconer’s formula is a simplified and outdated approach compared to SEM. Our intention, however, was not to conduct a formal comparison between family-based and SNP-based estimates. We used Falconer’s formula only as a supplementary “sanity check” which is why, as we also specify in the manuscript, it was not part of our preregistration. While the sensitivity analyses corroborated our main results, we felt it important to be transparent and report all analyses, which is why we presented them in their simplest form. In the revised “Genotype Data Processing and Relatedness Estimation” section (page 21), we now explicitly state that these estimates were derived using a traditional method and are not intended as advanced twin modelling:

For validation, we additionally calculated heritability using Falconer’s formula (see SI Methods). This traditional twin-based analysis was not preregistered and used only as a supplementary check.”

Major comment 4. Do you believe your sample had adequate power to detect GxE interaction effects?

We thank the reviewer for raising this important point. Statistical power, defined as the probability of correctly rejecting the null hypothesis, in our case reflects the ability to detect a moderation effect of SES on the A, C, and/or E components if such an effect truly exists. Indeed, for 3 of the 4 imaging metrics we found moderation effects either at global and/or lobar resolution. For chi-square-based tests, power can be determined by the effect size of interest (Cohen’s w), the degrees of freedom, the sample size, and the significance threshold. Larger effect sizes and samples increase power, whereas higher degrees of freedom or stricter significance thresholds reduce it.

Following the reviewer's suggestion, we conducted power analyses using the "pwr.chisq.test" function in R, testing small ($w = 0.10$), medium ($w = 0.30$), and large ($w = 0.50$) effects. With a sample size of 9080 individuals and 3 degrees of freedom, these analyses indicated essentially 100% power to detect effects of these magnitudes if present, with $\alpha = 0.05$. To be clear, as our actual analyses apply FDR correction (which is stricter than $\alpha = 0.05$), these values represent an upper bound on our power. To further probe sensitivity, we tested: given 80% power, what is the smallest effect we could detect? This analysis suggested that we could reliably detect effects as small as $w \approx 0.03$. Thus, one could speculate that if present, undetected effect of SES on the A C and E of sulcal depth for instance is likely to be very small. It is important to note, however, that statistical power depends on many other factors which were not covered in the chi-square power test, including the variance of the SES moderator. Taken together, our analyses suggests that we had strong power to detect SES moderation effects if present, implying that any non-detected true effects are likely to be very small.

Major comment 5. Was there an association between scanner site & SES (i.e., did some sites collect a greater proportion of lower SES participants)? If so, it would be nice to see some further evidence that these site effects were entirely removed through data harmonisation.

We appreciate this comment and agree that within the US, SES will naturally co-vary with each individual site (and scanner; see Reviewer Figure 5). Our aim was to minimize scanner-related biases, which, if left uncorrected, can strongly confound multi-site imaging analyses and affect both the mean and variance of imaging metrics. To address this, we performed adjustments at the level of each individual scanner ($n = 29$), as certain sites have several. While scanner effects can never be completely removed, the originally presented box plots (SI Figures 5–8) demonstrate a substantial reduction of scanner bias. Importantly, our goal was not to remove SES-related differences across scanner, as these reflect true population variation and are central to our study. For this reason, we explicitly included parental education, parental income, and neighborhood deprivation as covariates in the ComBat model to preserve such variation.

Reviewer Figure 5. Box plots of the median and spread of the SES-index within each of the 29 scanners (n = 9080).

In response to the reviewer's comment, we tested whether scanner explained residual variance in the harmonized global brain metrics by fitting linear models with scanner as the sole predictor. Scanner accounted for 2% of the variance in harmonized surface area, 1% in thickness and GWC, and 0% in sulcal depth (Reviewer Table 2), confirming that scanner effects were strongly minimized by neuroComBat.

Metric	R2 scanner
Thickness	0.01
Surface area	0.02
GWC	0.01
Sulcal depth	0.00

Reviewer Table 2. Proportion of variance (R^2) in harmonized global imaging metrics accounted for by scanner effects.

Minor comment 6. I'm uncertain what the difference is between h2 and A in the figure captions. A is additive genetic variance (but not expressed as a proportion of total phenotypic variance?).

We agree that the distinction between heritability and additive genetics was unclear in our original manuscript, and we confirm that the reviewer's understanding is correct. As noted in our detailed response to Reviewer 1 Minor comment 4, who raised a similar point, we have revised the manuscript to clarify this distinction. Based on the reviewer comment we have additionally clarified this within the figure legends as seen exemplified in response to Reviewer 1 minor comment 5.

Reviewer #4

I co-reviewed this manuscript with one of the reviewers who provided the listed reports. This is part of the Communications Biology initiative to facilitate training in peer review and to provide appropriate recognition for Early Career Researchers who co-review manuscripts.

We thank the reviewer for taking the time to give constructive comments that have made our manuscript better!

References

- Beck, D., Ferschmann, L., MacSweeney, N., Norbom, L. B., Wiker, T., Aksnes, E., Karl, V., Dégeilh, F., Holm, M., Mills, K. L., Andreassen, O. A., Agartz, I., Westlye, L. T., Von Soest, T., & Tamnes, C. K. (2023). Puberty differentially predicts brain maturation in male and female youth: A longitudinal ABCD Study. *Developmental Cognitive Neuroscience*, *61*, 101261. <https://doi.org/10.1016/j.dcn.2023.101261>
- Beck, D., Whitmore, L., MacSweeney, N., Briant, A., Karl, V., De Lange, A.-M. G., Westlye, L. T., Mills, K. L., & Tamnes, C. K. (2025). Dimensions of Early-Life Adversity Are Differentially Associated With Patterns of Delayed and Accelerated Brain Maturation. *Biological Psychiatry*, *97*(1), 64–72. <https://doi.org/10.1016/j.biopsych.2024.07.019>
- Bourdieu, P. (2011). The forms of capital.(1986). *Cultural Theory: An Anthology*, *1*(81–93), 949.
- Clark, D. A., Hicks, B. M., Angstadt, M., Rutherford, S., Taxali, A., Hyde, L., Weigard, A. S., Heitzeg, M. M., & Sripada, C. (2021). The General Factor of Psychopathology in the Adolescent Brain Cognitive Development (ABCD) Study: A Comparison of Alternative Modeling Approaches. *Clinical Psychological Science*, *9*(2), 169–182. <https://doi.org/10.1177/2167702620959317>
- Fan, C. C., Loughnan, R., Wilson, S., Hewitt, J. K., ABCD Genetic Working Group, Agrawal, A., Dowling, G., Garavan, H., LeBlanc, K., Neale, M., Friedman, N., Madden, P., Little, R., Brown, S. A., Jernigan, T., & Thompson, W. K. (2023). Genotype Data and Derived Genetic Instruments of Adolescent Brain Cognitive Development Study® for Better Understanding of Human Brain Development. *Behavior Genetics*, *53*(3), 159–168. <https://doi.org/10.1007/s10519-023-10143-0>
- Farah, M. J. (2017). The Neuroscience of Socioeconomic Status: Correlates, Causes, and Consequences. *Neuron*, *96*(1), 56–71. <https://doi.org/10.1016/j.neuron.2017.08.034>
- Fujishiro, K., Xu, J., & Gong, F. (2010). What does “occupation” represent as an indicator of socioeconomic status?: Exploring occupational prestige and health. *Social Science & Medicine*, *71*(12), 2100–2107. <https://doi.org/10.1016/j.socscimed.2010.09.026>
- Lindseth, L. R. S., Beck, D., Westlye, L. T., Tamnes, C. K., & Norbom, L. B. (2025). Linking pregnancy- and birth-related risk factors to a multivariate fusion of child cortical structure. *Proceedings of the National Academy of Sciences*, *122*(25), e2422281122. <https://doi.org/10.1073/pnas.2422281122>

- Liu, W. M., Ali, S. R., Soleck, G., Hopps, J., dunston, K., & Pickett, T. (2004). Using Social Class in Counseling Psychology Research. *Journal of Counseling Psychology*, *51*(1), 3–18. <https://doi.org/10.1037/0022-0167.51.1.3>
- Long, K., & Renbarger, R. (2023). Persistence of Poverty: How Measures of Socioeconomic Status Have Changed Over Time. *Educational Researcher*, *0013189X2211414*. <https://doi.org/10.3102/0013189X221141409>
- Norbom, L. B., Hanson, J., van der Meer, D., Ferschmann, L., Røysamb, E., von Soest, T., Andreassen, O. A., Agartz, I., Westlye, L. T., & Tamnes, C. K. (2022). Parental socioeconomic status is linked to cortical microstructure and language abilities in children and adolescents. *Developmental Cognitive Neuroscience*, *56*, 101132. <https://doi.org/10.1016/j.dcn.2022.101132>
- Norbom, L. B., Rokicki, J., Eilertsen, E. M., Wiker, T., Hanson, J., Dahl, A., Alnæs, D., Fernández-Cabello, S., Beck, D., Agartz, I., Andreassen, O. A., Westlye, L. T., & Tamnes, C. K. (2023). *Parental education and income are linked to offspring cortical brain structure and psychopathology at 9-11 years* [Preprint]. *Radiology and Imaging*. <https://doi.org/10.1101/2023.02.16.23286027>
- Pizzagalli, F., Auzias, G., Yang, Q., Mathias, S. R., Faskowitz, J., Boyd, J. D., Amini, A., Rivière, D., McMahon, K. L., De Zubicaray, G. I., Martin, N. G., Mangin, J.-F., Glahn, D. C., Blangero, J., Wright, M. J., Thompson, P. M., Kochunov, P., & Jahanshad, N. (2020). The reliability and heritability of cortical folds and their genetic correlations across hemispheres. *Communications Biology*, *3*(1), 510. <https://doi.org/10.1038/s42003-020-01163-1>
- Rakesh, D., Zalesky, A., & Whittle, S. (2021). Similar but distinct – Effects of different socioeconomic indicators on resting state functional connectivity: Findings from the Adolescent Brain Cognitive Development (ABCD) Study®. *Developmental Cognitive Neuroscience*, *51*, 101005. <https://doi.org/10.1016/j.dcn.2021.101005>
- Rakesh, D., Zalesky, A., & Whittle, S. (2022). Assessment of Parent Income and Education, Neighborhood Disadvantage, and Child Brain Structure. *JAMA Network Open*, *5*(8), e2226208. <https://doi.org/10.1001/jamanetworkopen.2022.26208>
- Williams, C. J. (1993). On the covariance between parameter estimates in models of twin data. *Biometrics*, *49*(2), 557–568.
- Yosso, T. (2005). Whose cultural has capital. *A Critical Race Theory Discussion Of*.